# Performance Analysis and Prediction of 5G Round-Trip Time Based on the VMD-LSTM Method

**DOI:** 10.3390/s24206542

**Published:** 2024-10-10

**Authors:** Sanying Zhu, Shutong Zhou, Liuquan Wang, Chenxin Zang, Yanqiang Liu, Qiang Liu

**Affiliations:** 1School of Mechanical Engineering and Automation, Beihang University, Beijing 100191, China; zsyyy1003@buaa.edu.cn (S.Z.); 18374384@buaa.edu.cn (S.Z.); liuquanwang@buaa.edu.cn (L.W.); zcx2017@buaa.edu.cn (C.Z.); 2Jiangxi Research Institute, Beihang University, Nanchang 330096, China; 3Research and Application Center of Advanced CNC Machining Technology, State Administration of Science, Technology and Industry for National Defense, Beijing 100191, China

**Keywords:** 5G round-trip time prediction, real factory scenario, low latency requirements, stationary analysis, spectrum analysis, variational mode decomposition, long short-term memory

## Abstract

With the increasing level of industrial informatization, massive industrial data require real-time and high-fidelity wireless transmission. Although some industrial wireless network protocols have been designed over the last few decades, most of them have limited coverage and narrow bandwidth. They cannot always ensure the certainty of information transmission, making it especially difficult to meet the requirements of low latency in industrial manufacturing fields. The 5G technology is characterized by a high transmission rate and low latency; therefore, it has good prospects in industrial applications. To apply 5G technology to factory environments with low latency requirements for data transmission, in this study, we analyze the statistical performance of the round-trip time (RTT) in a 5G-R15 communication system. The results indicate that the average value of 5G RTT is about 11 ms, which is less than the 25 ms of WIA-FA. We then consider 5G RTT data as a group of time series, utilizing the augmented Dickey–Fuller (ADF) test method to analyze the stability of the RTT data. We conclude that the RTT data are non-stationary. Therefore, firstly, the original 5G RTT series are subjected to first-order differencing to obtain differential sequences with stronger stationarity. Then, a time series analysis-based variational mode decomposition–long short-term memory (VMD-LSTM) method is proposed to separately predict each differential sequence. Finally, the predicted results are subjected to inverse difference to obtain the predicted value of 5G RTT, and a predictive error of 4.481% indicates that the method performs better than LSTM and other methods. The prediction results could be used to evaluate network performance based on business requirements, reduce the impact of instruction packet loss, and improve the robustness of control algorithms. The proposed early warning accuracy metrics for control issues can also be used to indicate when to retrain the model and to indicate the setting of the control cycle. The field of industrial control, especially in the manufacturing industry, which requires low latency, will benefit from this analysis. It should be noted that the above analysis and prediction methods are also applicable to the R16 and R17 versions.

## 1. Introduction

In the era of Industry 4.0 [1,2,3], the industrial sector urgently needs to introduce transmission protocols characterized by high transmission rates, low latency, and high reliability [4] to meet the demand for massive data transmission in factories [1]. Unlike wired protocols, wireless networks better meet the current industrial demand for device mobility, leading researchers to explore their applications in the sector [5,6,7,8,9] and propose standards such as industrial wireless sensor networks [5] and WIA-FA for factory automation [6], which feature lower transmission latency. In these applications, WIA-FA has achieved the lowest transmission delay of 25 ms in multi-AGV synchronous motion scenarios. However, most of these wireless networks have limited coverage and narrow bandwidth, making it difficult to ensure the determinism of information transmission.

The commercially available 5G network is renowned for its low latency and high reliability, offering broad application prospects [10]. The current state-of-the-art 6G network outperforms 5G in terms of real-time performance and reliability, but it is still in the theoretical research phase. Ref. [11] proposed a dual-function radar-communication base station beamforming design method to address joint communication and signal optimization issues in the 6G environment. Ref. [12] identified 12 key scientific challenges facing 6G. Overall, 6G technology is still far from practical application. Against this backdrop, researching the application potential of 5G in the industrial sector is particularly necessary.

The measurement and performance analysis of 5G RTT data can provide a basis for assessing its application potential in specific industrial scenarios. Therefore, selecting appropriate 5G RTT measurement methods based on different scenario requirements is crucial. References [13,14] focus on the characteristics of the network itself, using precise but complex measurement tools or methods to analyze 5G RTT under different network conditions. However, these methods are not suitable for engineering practice due to their lack of convenience. Among researchers focusing on industrial applications, the author [15] conducted 5G RTT measurements and characteristic analyses on highways. Ref. [16] improved urban positioning accuracy by measuring 5G RTT in the Global Navigation Satellite System (GNSS). Although these findings demonstrate the potential of 5G, there is a lack of research on its RTT performance in real factory environments.

In industrial fields with high real-time requirements, accurately predicting 5G RTT helps improve system performance. For instance, in scenarios involving dual AGV (Kunshan Huaheng Welding Co., Ltd., Kunshan, China) synchronized control, precise 5G RTT predictions assist in maintaining the accuracy and stability of the synchronized control system [17]. The author of [18] utilized the specialized measurement software Nemo Handy (Keysight Technologies, Inc., Santa Rosa, CA, USA) to obtain real-time data on radio layer and physical layer parameters and combined it with machine learning algorithms to predict 5G RTT. The author of [19] proposed a URLLC time delay prediction method based on unbalanced regression algorithms and LSTM to address sporadic large time delay issues. These studies demonstrate the effectiveness of 5G prediction methods within their respective research contexts, but there is a lack of research on 5G time delay prediction in real factory environments.

To address the research gap in existing literature regarding the application of 5G in factory environments, we designed a 5G RTT measurement method suitable for factories, analyzed the measured RTT performance, and proposed a method for predicting 5G RTT. Our study confirms the feasibility of applying 5G in factory environments with low latency requirements, which is of significant importance in the current context of Industry 4.0 and provides a valuable reference for future exploration of 6G industrial applications.

The rest of the paper is organized as follows. Section 2 briefly provides an overview of research works related to this study. Section 3 details the test environment setting for 5G network performance. In Section 4, statistical analysis is conducted on the 5G RTT data, and the results indicate that the low latency performance of 5G technology is superior to existing standards. At the same time, the statistical results provide a basis for assessing whether 5G networks can be applied to specific industrial scenarios. Section 5 demonstrates that the RTT data are a series of non-stationary random time sequences. In Section 6, first-order differencing is first used on the original 5G RTT series to obtain the stationary differential sequences. Then, a time series analysis-based variational mode decomposition (VMD)–long short-term memory (LSTM) prediction method is proposed, in which VMD is used to decompose the differential sequences into a series of different modes to reduce the impact of the complexity and volatility of the original data on prediction accuracy. After that, LSTM is used to predict the differential sequences, and finally, the inverse difference is performed to obtain the predicted value of 5G RTT. Section 7 first introduces two metrics for evaluating model prediction performance based on the accuracy and stability requirements of industrial control systems. Then, a sensitivity analysis of the model to different hyperparameters is conducted, providing a basis for hyperparameter selection. Finally, by comparing different prediction methods, it proves that the proposed method has a better performance in the prediction of RTT. Finally, conclusions are presented in Section 8.

## 2. Related Works

### 2.1. Measurement and Analysis of 5G RTT

To determine whether 5G can meet the requirements of applications in real factory scenarios, as described in this paper, it is necessary to measure 5G RTT to analyze its performance. The authors of [15] investigated the latency and RTT characteristics of a live 5G NSA network, but the experimental environment was conducted on highways rather than in factories. The authors of [13] used three locally hosted measurement methods to present initial measurements of the data rate and the round-trip delay in standalone NPNs with various end devices, but the experimental environment was conducted on highways rather than in factories. The authors of [14] measured the one-way packet delays with sub-microsecond precision as well as measuring the packet core delay with nanosecond precision, yet in the field of industrial control, the measurement accuracy of RTT should be consistent with the control cycle.

To meet the requirement of measuring 5G RTT in factory scenarios while avoiding the waste of time and manpower, we utilized commonly used control components in factories, such as PLCs, and other commercial network devices as our experimental equipment. We employed a simple Ping command to record response times. Through our measurements, we obtained RTT data from the actual factory scenario and provided a range of 5G performance, which serves as a reference for determining whether 5G can be selected as the network communication protocol for control scenarios.

### 2.2. Prediction of 5G RTT

After determining the performance of 5G, we predict the RTT of 5G. The current prediction methods mainly include models based on 5G RTT theory and methods based on 5G RTT timing.

#### 2.2.1. Models Based on 5G RTT Theory

The model based on 5G RTT theory is a theoretical analysis method. It establishes a network environment model using mathematical methods based on the structure of the communication system and the principles of signal propagation. Due to the high computational complexity of complex systems, some scholars use computer simulations [20,21] to obtain RTT. However, when the network environment is complex, it is very labor-intensive and prone to errors, such as the actual factory scenario in this paper. 

#### 2.2.2. Methods Based on 5G RTT Timing

Before predicting time series, it is necessary to extract their features. Simple feature extraction methods include the Euclidean Distance (ED) [22] and Dynamic Time Warping (DTW) [23], but they are sensitive to outliers in the data. Among the commonly used methods currently, the symbolization methods for time series mainly include piecewise aggregate approximation (PAA), symbolic aggregate approximation (SAX) [24], and singular value decomposition (SVD) [25], but it is difficult to measure the similarity of symbols. Typical time-frequency analysis methods include Fourier transform (FT) [26] for stationary sequences and Wavelet transform (WT) [27] for non-stationary sequences. To solve the difficulty of wavelet base selection, empirical mode decomposition (EMD) and ensemble EMD (EEMD) [28,29] are applied. To overcome modal aliasing and enhance the adaptability and flexibility of the algorithm, we have chosen the more effective variational mode decomposition (VMD) [19,30,31,32,33,34] method for feature extraction from the 5G RTT series.

After extracting the time series features, we make predictions. Among the model-based time series predicting methods, representative ones include the hidden Markov model (HMM) [35], AR [36], and its derived models, such as autoregressive moving average (ARMA) [37], autoregressive integrated moving average (ARIMA) [38], and vector autoregression (VAR) [39], but they can only be applied to stationary time series [40]. Based on the analysis of the 5G RTT series presented in this paper, they exhibit non-stationarity. Therefore, instead of adopting model-based methods to predict 5G RTT, we have chosen artificial intelligence methods. Artificial neural networks (RNNs) [41] are well suited for time series forecasting tasks due to their memory capabilities. However, RNNs can only retain short-term memory [42], which is not suitable for the time series used in this study. Based on the RNN, a model called long short-term memory (LSTM) [43] is improved with a long-term memory function developed by introducing the concept of cell state [19,28,30,31,32,33,44], which has a better effect than an RNN in long-sequence forecasts. As a variant of the LSTM algorithm, gated recurrent unit (GRU) has a faster training speed, but its prediction is not as accurate as LSTM [45]. Especially for predictions with a longer time span, LSTM has a significant accuracy advantage [46]. Although the TCN proposed in recent years has a faster calculation speed, its prediction accuracy is not satisfactory [47]. Furthermore, our partial autocorrelation analysis indicates that 5G time series exhibit long-term autocorrelation, making LSTM a suitable choice for prediction. Therefore, we have selected LSTM as the prediction method.

#### 2.2.3. Examples of 5G RTT Prediction

Currently, some scholars [28] have researched predicting 5G RTT performance, and [31,32,34] have adopted prediction methods similar to those in this paper. However, their application scenarios or data sources are not from real factories. The authors of [19] proposed a URLLC time delay prediction method but focused on predicting occasional large time delays. The authors of [30] presented a 5G-A terminal transmission delay prediction algorithm but did not provide details of the network testbed and evaluation of the method’s performance.

The research in this paper focuses on conducting RTT measurements and predictions for commercial network services targeted at factory scenarios and providing assistance for improving process control performance. Therefore, after reviewing the current relevant research findings, we have selected hardware, software, and analytical methods that align with the research content of this paper.

### 2.3. The Contributions of This Paper

The contributions of this paper are as follows:For real factory scenarios, we propose a 5G RTT prediction method with low prediction error and good transferability, demonstrating the feasibility of applying 5G in factory scenarios with low latency requirements;The sensitivity analysis of the model’s prediction performance to parameters provides readers with a basis for selecting model parameters;The proposed model prediction accuracy metric *TC*, combined with the control domain, offers a new perspective on how to select control periods and when to retrain the model.

## 3. Test Environment Setting

Internet of Things (IoT) devices in smart factories include machine tools, robotic arms, etc., and programmable logic controllers (PLCs), which are widely used as their data acquisition and control components. Thus, to analyze the RTT characteristics of a 5G network in a real factory scenario, we chose a soft PLC (Kunshan Huaheng Welding Co., Ltd., Jiangsu, China) (including a master PLC and slave PLC) as the experimental equipment and constructed a 5G test network for RTT testing. The 5G version used in the network was standard R15 [48]. The base station we used was 5G LampSite (Huawei Technologies Co., Ltd., Guangdong, China). The transceivers were Aumiwalker’s R511 (Aumiwalker Technology Co., Ltd., Beijing, China) [49], and other devices in the network were also existing commercial products rather than prototypes. The structure of the network is shown in Figure 1.

A network layer tunnel was established through a 5G local area network (LAN) and communicated with transceivers. The master PLC (MPLC) can be deployed after the internal mobile edge computing (MEC) of the enterprise or on the end side. The slave PLC (SPLC) was on the base station side. For the physical layer technique, some researchers have elaborated on the important impact of channel coding and modulation schemes on the reliability and efficiency of the transmission: The authors of [50] designed an SSERGMS constellation and P-LDPC code to remarkably improve the convergence and decoding performance of MIMO-VLC systems. The authors of [51] compared the characteristics and advantages of three commonly used codes (Turbo, LDPC, and Polar), as well as their application-specific integrated circuit (ASIC) implementation of the decoders, to guide the selection and design of channel coding. Although the focus of this study is not on optimizing coding methods or modulation schemes, it was still necessary to specify the settings of the relevant items in the test network: LDPC code was used for the service channel, and Polar code was used for the control channel. Using adaptive modulation and coding (AMC) to select an appropriate modulation scheme based on the current channel quality, the modulation indices corresponding to different modulation schemes are shown in Table 1.

Table 2 summarizes the system configuration of the test network.

In the 5G RTT test process, we pinged a packet from the SPLC to the MPLC, and the RTT value was obtained by recording the time of sending and receiving the packet. The end-to-end ping delay included two parts: the air interface delay and the delay from the base station to the server. We used different packet sending intervals and package lengths in each group of tests while the number of packets remained at 50,000. Table 3 shows the packet settings and test results of the 5G RTT.

## 4. Statistical Analysis of 5G RTT Data

According to different packet sending intervals and packet lengths, eight RTT datasets (dataset 1 to dataset 8 in Table 3) were statistically analyzed, and the results are reported in Table 3. To apply 5G networks to industrial scenarios, it is necessary to first determine the required network service capabilities based on business requirements and then plan a 5G network that can achieve such network service capabilities. There are already relevant standardized regulations for network planning. Therefore, for 5G networks in different industrial scenarios, as long as they have the same network service capabilities as the network used in our research, the characteristics of the network after standardized network planning will be approximately the same. Consequently, our research results can be generalized for other industrial scenarios with the same business requirements.

It can be seen that the minimum values of the eight sets of data were relatively close and all less than 7 ms. Due to the uncertainty caused by factors such as the network signal-to-noise ratio (SNR) and network load, the maximum RTT had a strong randomness, yet the average values of the eight sets of data were about 11 ms. Generally speaking, when the packet lengths are the same, the longer the packet sending intervals, the larger the average values of RTT, yet the increase is relatively small. It can be seen from the standard deviation data that when the packet sending intervals increased while the packet lengths were the same, the variability in the 5G RTT increased. All eight sets of data had positive skewness because of the long tailing and the large number of great RTT data, and this situation was influenced by the SNR as well as the load of the network.

We used dataset 1 as an example to draw its histogram and probability density function (PDF), and the results are presented in Figure 2a, while Figure 2b displays its cumulative histogram.

According to the statistical results, the RTT data with values within the range of (7, 14] accounted for the vast majority of the total sample, reaching 87.8%, yet the RTT data with a value greater than 15 ms only accounted for 3.6%. In addition, the RTT data with a value greater than 20 ms (which is regarded as a large delay) were even less, accounting for only 0.3%.

Through the statistical analysis of 5G RTT data in this section, it can be seen that their average value was around 11 ms, which is better than the 25 ms of WIA-FA and can better meet the low latency requirements of machine tools or robot control in factory environments.

## 5. Time Series Analysis of 5G RTT Data

The 5G RTT data can be regarded as a time series, and the 5G RTT series can be represented as {Rt,t=1,…,TN}. Here, Rt∈ℝ+ and TN∈ℤ+, Rt is the RTT value at time *t*, and TN is the length of the RTT series. In the following text, we simplify the mathematical expression of the 5G RTT series to {Rt}.

In this section, stationary analysis and correlation coefficient analyses are conducted on $\left\{R_{t}\right\}$, and the results are used in the VMD-LSTM method proposed in the next section.

### 5.1. Stationary Analysis of RTT Series

We used the ADF test [52] to analyze the stationarity of the RTT series. Due to the long length of the original RTT series, we divided it into $\left\lceil RLen/s\right\rceil$ segments of short sequences and conducted stationarity tests on the short sequences. Among them, *RLen* represents the length of the original sequence. According to Section 3, *Rlen* = 50,000. Symbol *s* represents the length of the short sequence. Because the errors in the ADF test may affect the results of the stationarity test when the sequence length is short, we took *s* = 1000 as an example to perform the ADF test on short RTT sequences. Due to the sensitivity of the ADF test results to the lag length *lp*, we set the lag length *lp* of the short sequence to a value where the absolute value of the τ-statistic is greater than 1.6, based on the research results of Ng and Perron [53]. We listed the ADF test results of several short RTT sequences in Table 4, where the critical value (0.05) denotes the corresponding critical value when the significance is equal to 0.05. To ascertain the stationarity of a short RTT sequence, we applied the following criterion: if the absolute value of the τ-statistic was greater than the critical value (0.05) and the *p*-value was less than the significance level (0.05) set in this study, then the short RTT sequence was considered stationary.

The *p*-value, τ-statistic, and critical values are shown in Table 4.

None of the five short sequences shown in Table 4 satisfied the above rule, so they were all non-stationary. As a supplement, we also conducted ADF tests on the short sequences with a length of *s* = 2000:1000:10,000, and the results indicated that, regardless of the value of sequence length, some or all of the sequence segments were non-stationary. Therefore, it can be concluded that the 5G RTT series are non-stationary.

Due to the poor accuracy of predicting non-stationary 5G RTT series directly, we constructed a first-order differential sequence $\left\{DR_{t},t=2,\ldots ,T_{N}\right\}$ of the RTT series, and its calculation formula is
(1)$$DR_{t}=R_{t}-R_{t-1}$$

$\left\{DR_{t}\right\}$ was also divided according to the length of the sequence segment of 1000. Since the length of the first-order differential sequence was 49,999, we could obtain 49 short sequences with a length of 1000 and 1 short sequence with a length of 999. The above ADF test process was performed on each short sequence. The test results indicated that the first-order differential sequence of RTT exhibited stationarity at a significance level of 0.05. In the following text, we will abbreviate the first-order differential RTT as DRTT.

### 5.2. Correlation Coefficient Analysis of Differential RTT Series

We calculated the autocorrelation coefficient and partial autocorrelation coefficient for the DRTT series $\left\{DR_{t}\right\}$ with different sequence lengths. The results showed the autocorrelation coefficients and partial autocorrelation coefficients of these sequence segments had similar characteristics. Therefore, only the results of the autocorrelation coefficients and partial autocorrelation coefficients of one segment of the short DRTT sequence with a length of 1000 were plotted separately. In Figure 3a,b, the dashed line marks two times the standard deviation, and the bars are color-coded to indicate if the correlation coefficients fall within this range.

Figure 3a shows that the autocorrelation coefficient of the short DRTT sequence decayed extremely slowly and only decayed to within the range of two times the standard deviation after 850 lags. Since the ADF test results indicated the stationarity of the DRTT series, and according to the Wold decomposition theorem [54], i.e., a discrete stationary sequence can be decomposed into a deterministic sequence and a stochastic sequence, we knew the random components in the DRTT series had a significant impact on the sequence, resulting in complex nonlinear characteristics. Figure 3b shows that after the 40th lag, 90% of the partial autocorrelation coefficients were within the range of two times the standard deviation. Therefore, the partial autocorrelation coefficient can be approximately considered as having a 40th-order truncation, which means that the DRTT value $DR_{t}$ at time *t* has a high correlation with the previous forty DRTT values $DR_{t-40},\ldots ,DR_{t-1}$. In the subsequent prediction process, it was proven that adjusting the parameters of our RTT prediction method by combining the correlation coefficients of the sequence played an important role in improving prediction accuracy.

For various time series prediction methods, whether traditional or emerging, the prediction performance of nonlinear sequences is worse than that of linear sequences. Therefore, to improve the prediction accuracy of 5G RTT, a method based on VMD-LSTM was proposed in this study, and the first-order differential sequence of the RTT series was utilized as the input of this prediction method.

## 6. 5G RTT Prediction with the Time Series Analysis-Based VMD-LSTM Method

We propose a 5G RTT prediction method based on time series analysis and VMD-LSTM, to use a sequence of *m* + 1 RTT data ${\{R}_{t,-m}\}=\{R_{t-m-1},\ldots ,R_{t-1}\}$ obtained through measurement to predict a sequence of *n* RTTs ${\{R}_{t,n}\}=\{R_{t},\ldots ,R_{t+n-1}\}$ that will be generated in the future. $\{DR_{t}^{k}\},k=1,\ldots ,K$ represents the *k*th subsequence of the *K* subsequences obtained after the VMD process conducted on $\left\{DR_{t}\right\}$. In addition, the terms related to LSTM networks are defined as follows: $LSTM_{t-i,k}(i=1,\ldots ,m;k=1,\ldots ,K)$ represents the LSTM unit at time *t* − *i* in the LSTM network corresponding to the *k*th subsequence. Among them, $C_{t-i-1,k}$ represents the cell state of the LSTM unit at time *t* − *i* − 1, $h_{t-i-1,k}$ represents the output of the LSTM unit at time *t* − *i* − 1, $DR_{t-i}^{k}$ represents the *k*th component of the known DRTT data input to the LSTM unit at time *t* − *i*, $C_{t-i,k}$ represents the cell state of the LSTM unit at time *t* − *i*, $h_{t-i,k}$ represents the output of the LSTM unit at time *t* − *i*. These LSTM units ($LSTM_{t-i,k}$) make up the training phase of the LSTM network. $LSTM_{t+j,k}(while:n\geq 2,j=0,\ldots ,n-2,while:n=1,j=-1;k=1,\ldots ,K)$ represents the LSTM unit at time *t* + *j* in the LSTM network corresponding to the *k*th subsequence. Among them, $C_{t+j-1,k}$ represents the cell state of the LSTM unit at time *t + j* − 1, $h_{t+j-1,k}$ represents the output of the LSTM unit at time *t + j* − 1, $DR_{t+j}^{k}$ represents the *k*th component of the predicted DRTT data input to the LSTM unit at time *t* + *j*, $C_{t+j,k}$ represents the cell state of the LSTM unit at time *t* + *j*, $h_{t+j,k}$ represents the output of the LSTM unit at time *t* + *j*, i.e., the predicted DRTT data for the next moment (time *t* + *j* + 1). These LSTM units ($LSTM_{t+j,k}$) make up the prediction phase of the LSTM network. In the framework diagram shown in Figure 4, it can be seen that the four main steps of this time series analysis-based VMD-LSTM method are as follows:

Step 1. First-order differencing is performed on the 5G RTT series $\left\{R_{t,-m}\right\}$ to obtain the DRTT series $\left\{DR_{t,-m}\}=\{DR_{t-m},\ldots ,DR_{t-1}\right\}$, where $DR_{t}$ is calculated according to (1).

Step 2. VMD is performed on the 5G DRTT series.

According to the analysis in Section 5, it can be concluded that due to the nonlinearity of the 5G DRTT series, the accuracy of directly predicting DRTT data is relatively low. Therefore, in the prediction method proposed in this paper, the VMD method is first used to decompose the DRTT series $\left\{DR_{t,-m}\right\}$ into *K* subsequences $\{DR_{t,-m}^{k}\},k=1,\ldots ,K$ and a residual sequence $\left\{DR_{t,-m}^{res}\right\}$, where $\left\{DR_{t,-m}^{k}\}=\{DR_{t-m}^{k},\ldots ,DR_{t-1}^{k}\right\}$ and $\left\{DR_{t,-m}^{res}\}=\{DR_{t-m}^{res},\ldots ,DR_{t-1}^{res}\right\}$. Besides, $\left\{DR_{t,-m}\}=\sum_{k=1}^{K}\left\{DR_{t,-m}^{k}\right\}+\{DR_{t,-m}^{res}\right\}$ and $DR_{t-x}=\sum_{k=1}^{K}DR_{t-x}^{k}+DR_{t-x}^{res},x=1,2,\ldots ,m$.

Step 3. LSTM prediction is performed separately on the decomposed *K* + 1 subsequences.

After completing step 2, we obtain *K* IMFs and 1 residual sequence, totaling *K* + 1 subsequences. As these subsequences possess different frequency domain characteristics, it is necessary to establish individual LSTM networks for each of the *K* + 1 subsequences before making predictions on them separately. Subsequently, these *K* + 1 LSTM networks will be used for predicting the corresponding subsequences, respectively.

Taking the first subsequence $\left\{DR_{t,-m}^{1}\right\}$ as an example, the prediction process can be described as follows: by substituting *m* DRTT values, i.e., $DR_{t-m}^{1},\ldots ,DR_{t-1}^{1}$ in $\left\{DR_{t,-m}^{1}\}=\{DR_{t-m}^{1},\ldots ,DR_{t-1}^{1}\right\}$, into the already modelled network LSTM_1_, the predicted values $D{\hat{R}}_{t}^{1},\ldots ,D{\hat{R}}_{t+n-1}^{1}$ for the next *n* unknown DRTT values $DR_{t}^{1},\ldots ,DR_{t+n-1}^{1}$ can be calculated.

Define $\left\{D{\hat{R}}_{t,n}^{1}\}=\{D{\hat{R}}_{t}^{1},\ldots ,D{\hat{R}}_{t+n-1}^{1}\right\}$ as the predicted sequence of the first subsequence. For other *K* subsequences, their respective LSTM networks can be used to calculate *n* predicted values. These predicted values respectively constitute the prediction sequences $\left\{D{\hat{R}}_{t,n}^{2}\},\ldots ,\{D{\hat{R}}_{t,n}^{K}\right\}$ and $\left\{D{\hat{R}}_{t,n}^{res}\right\}$. Among them, $\left\{D{\hat{R}}_{t,n}^{1}\},\ldots ,\{D{\hat{R}}_{t,n}^{K}\right\}$ represents the predicted sequence of *K* IMFs, while $\left\{D{\hat{R}}_{t,n}^{res}\right\}$ represents the predicted sequence of the residual sequence.

Step 4. The prediction results of the original sequence $\left\{DR_{t,-m}\right\}$ are reconstructed.

After steps 1–3, we obtain the prediction results $\left\{D{\hat{R}}_{t,n}^{1}\},\ldots ,\{D{\hat{R}}_{t,n}^{K}\right\}$ and $\left\{D{\hat{R}}_{t,n}^{res}\right\}$ for each IMF and residual sequence of ${\{DR}_{t,-m}\}$. As described in step 2, the DRTT value at a certain moment is equal to the sum of the values of each IMF and the residual value at that moment, i.e., $DR_{t-x}=\sum_{k=1}^{K}DR_{t-x}^{k}+DR_{t-x}^{res},x=1,2,\ldots ,m$. Thus, the predicted DRTT value at a certain time in the future is also equal to the predicted values of each IMF plus the predicted residual value at that time, i.e., $D{\hat{R}}_{t+y-1}=\sum_{k=1}^{K}D{\hat{R}}_{t+y-1}^{k}+D{\hat{R}}_{t+y-1}^{res},y=1,2,\ldots ,n$. Therefore, the predicted sequence of the DRTT series can be represented by the following formula:(2)$$\left\{D{\hat{R}}_{t,n}\}=(\sum_{k=1}^{K}\{D{\hat{R}}_{t,n}^{k}\})+\{D{\hat{R}}_{t,n}^{res}\right\}$$

As mentioned earlier, the unknown sequence ${\{DR}_{t,n}\}=\{DR_{t},\ldots ,{DR}_{t+n-1}\}$ can be predicted using the known sequence ${\{DR}_{t,-m}\}=\{DR_{t-m},\ldots ,{DR}_{t-1}\}$, and the predicted sequence composed of predicted values is $\{D{\hat{R}}_{t,n}\}=\left\{D{\hat{R}}_{t},\ldots ,D{\hat{R}}_{t+n-1}\right\}$. By performing an inverse operation on (1), we can obtain:(3)$$\left\{\begin{array}{l} R_{t}=DR_{t}+R_{t-1}\\ R_{t+1}=DR_{t+1}+R_{t}=DR_{t+1}+DR_{t}+R_{t-1}\\ \cdots \\ R_{t+n-1}=DR_{t+n-1}+R_{t+n-2}=\sum_{j=t}^{t+n-1}DR_{j}+R_{t-1} \end{array}\right.$$

Meanwhile, as mentioned in step 1, the known sequence ${\{DR}_{t,-m}\}=\{DR_{t-m},\ldots ,{DR}_{t-1}\}$ is obtained using the first-order difference of the known sequence ${\{R}_{t,-m}\}=\{R_{t-m-1},\ldots ,R_{t-1}\}$. Therefore, in (3), only $R_{t-1}$ is known. So, we incorporate the predicted values of $R_{t-1}$ and the corresponding first-order difference of RTT into (3) and rewrite it as:(4)$$\left\{\begin{array}{l} {\hat{R}}_{t}=D{\hat{R}}_{t}+R_{t-1}\\ {\hat{R}}_{t+1}=D{\hat{R}}_{t+1}+{\hat{R}}_{t}=D{\hat{R}}_{t+1}+D{\hat{R}}_{t}+R_{t-1}\\ \cdots \\ {\hat{R}}_{t+n-1}=D{\hat{R}}_{t+n-1}+{\hat{R}}_{t+n-2}=\sum_{j=t}^{t+n-1}D{\hat{R}}_{j}+R_{t-1} \end{array}\right.$$

By using (4), the predicted values of *n* future RTTs can be calculated.

### 6.1. Decomposing the 5G DRTT Series with the VMD Method

VMD is a method that utilizes an iterative search for the optimal solution of a variational model to extract the intrinsic mode function (IMF) components of a signal. It is suitable for decomposing time series with nonlinear and non-stationary characteristics [33]. Using the VMD method, the first-order differential sequence $\left\{{DR}_{t}\right\}$ of the 5G RTT series can be decomposed into *K* modes $\left\{DR_{t}^{k}\right\},k=1,\ldots ,K$. The principle is to find a set of subsequences, under the condition that the sum of each subsequence is equal to the original sequence so that the sum of the estimated bandwidth of the subsequences is minimized [33]:(5)$$\left\{\begin{array}{c} \underset{\left\{DR_{t}^{k}\},\{\omega_{k}\right\}}{\min}\left\{{\sum_{k=1}^{K}\left\|\partial_{t}[(\delta_{t}+\frac{j}{\pi t})*DR_{t}^{k}]e^{-j\omega_{k}t}\right\|}_{2}^{2}\right\}\\ s.t.\sum_{k=1}^{K}DR_{t}^{k}=DR_{t} \end{array}\right.$$
in which $DR_{t}$ represents the *t*th element of the DRTT series, $DR_{t}^{k}$ represents the *t*th element in the *k*th DRTT subsequence, $\omega_{k}$ represents the center frequency of the *k*th DRTT subsequence. The symbol * represents convolution.

The problem can be transformed into the following variational problem using the Lagrange equation:(6)$$L\left(\left\{DR_{t}^{k}\right\},\left\{\omega_{k}\right\},\lambda \right)=\alpha {\sum_{k=1}^{K}\left\|\partial_{t}[(\delta_{t}+\frac{j}{\pi t})*DR_{t}^{k}]e^{-j\omega_{k}t}\right\|}_{2}^{2}+{\left\|DR_{t}-\sum_{k=1}^{K}DR_{t}^{k}\right\|}_{2}^{2}+<\lambda (t),DR_{t}-\sum_{k=1}^{K}DR_{t}^{k}>$$

In which λ represents a vector composed of Lagrange multipliers, α represents the quadratic penalty factor. Iteratively calculate the value $DR^{k,l}\left(\omega \right)$ in the frequency domain of the *k*th DRTT subsequence, as well as the corresponding center frequency of the *k*th differential subsequence $\omega_{k}^{l}$ and the Lagrange multipliers $\lambda^{l}$, through the process shown in Figure 5. Among them, *l* represents the number of iterations. When the convergence condition $\sum_{k}{\left\|DR^{k,l+1}(\omega )-DR^{k,l}(\omega )\right\|}_{2}^{2}/{\left\|DR^{k,l}(\omega )\right\|}_{2}^{2}<\varepsilon$ is met, the iteration process can be stopped, and *K* DRTT subsequences $\left\{DR_{t}^{k}\right\},k=1,\ldots ,K$ in the time domain can be obtained through inverse Fourier transform.

In the above decomposition process, the number of DRTT subsequences *K*, the quadratic penalty factor α, and the update coefficient τ have a significant impact on the decomposition effect. Among them, τ is used to update the Lagrange multiplier vector in the iterative calculation shown in Figure 5.

We referred to the method in [19] and set the value of *K* to 7. Since the optimization process for the quadratic penalty factor α and the update coefficient τ in [24] adopted the GOA method [55], which optimizing the mean of the residual, i.e., *REI* shown in (7) to minimum, this will cause the VMD to focus on obtaining the minimum residual while ignoring the consideration of the stationarity or randomness of the IMFs, which has a significant impact on prediction accuracy.
(7)$$REI=\frac{1}{LN}\sum_{t}DR_{t}^{res}$$
in which *LN* represents the length of the residual sequence.

To simplify the optimization process for α and τ while improving the predictability of the sequence, we propose a parameter optimization method based on correlation coefficients. As stated in Section 5.2, when the partial autocorrelation coefficient of the sequence has an *m*-order truncation, it indicates that the DRTT value at time *t* (i.e., $DR_{t}$) has a high correlation with the DRTT values at *m* previous moments (i.e., $DR_{t-m},\ldots ,DR_{t-1}$), while the correlation with the DRTT values at other previous moments is weak. Based on this feature, the more accurate the relationship between $DR_{t}$ and $DR_{t-m},\ldots ,DR_{t-1}$ we establish is, the closer the predicted value $D{\hat{R}}_{t}$ based on this relationship will be to the true value $DR_{t}$. Therefore, when conducting VMD, our optimization goal is to search for a set of α and τ so that the partial autocorrelation coefficients of each subsequence conform to the characteristic of *m*-order truncation as closely as possible. The steps are as follows:

Step 1. Initial values of 3000 and −0.1 are assigned to α and τ, respectively.

Step 2. VMD is executed, and the partial autocorrelation coefficients of *K* IMFs and residual sequences are respectively calculated, which form a matrix PA with $l_{e}$ rows and *K* + 1 columns. $l_{e}$ represents the maximum lag of the partial autocorrelation coefficient to be calculated. The element $PA_{i,j},i=1,\ldots ,le,j=1,\ldots ,K+1$ of the matrix represents the partial autocorrelation coefficient of the *j*th subsequence with *i*th lag.

Step 3. Search for whether there exists a lag *m* that all subsequences satisfy the following conditions:(8)$$\left\{\begin{array}{l} PA_{k}^{sum}\leq \theta_{1}\\ \forall PA_{k}\in PA_{k},PA_{k}\leq \theta_{2} \end{array}\right.$$

The calculation formulas for $PA_{k}^{sum}$, $\theta_{1}$, and $\theta_{2}$ are as follows:$$\left\{\begin{array}{l} PA_{k}^{sum}=\sum_{i=m+1}^{l_{e}}PA_{i,k}\\ \theta_{1}=\left(l_{e}-m\right)*\left(pb+\theta \right)\\ PA_{k}=\left[PA_{m+1,k},\ldots ,PA_{l_{e},k}\right] \end{array}\right.$$

Among them, *k* = 1, …, *K* represents the *k*th subsequence, pb is two times the standard deviation for the correlation analysis, while θ, $\theta_{1}$, and $\theta_{2}$ limit the degree of deviation of the partial autocorrelation coefficients of the subsequence from the *m*-order truncation. θ and $\theta_{1}$ constrain the sum of partial autocorrelation coefficients, and $\theta_{2}$ constrains the individual partial autocorrelation coefficients.

Step 4. If the value of *m* that meets the condition cannot be found in step 3, set τ=τ+0.01 and return to step 2 to restart the search. If the value of *m* cannot be found until τ=0.1, proceed to step 5.

Step 5. Let α=α−100, τ=−0.1 and return to step 2 to restart the search.

After optimizing the VMD parameters mentioned above, we have obtained a set of subsequences with good predictability. After calculation, the value of *m* is 82 for the chosen DRTT series. We plot the partial autocorrelation coefficient of one of the subsequences in Figure 6. It can be seen that after approximately 82nd lag, the value of the partial autocorrelation coefficient is relatively small, which can be roughly considered to have 82nd lag truncation characteristics.

### 6.2. 5G RTT Prediction Method Based on VMD-LSTM

LSTM is a special type of recurrent neural network (RNN) that not only takes the state of the previous moment as the input for the current moment but also takes the state of the past *m* moments as the input for the current moment. Therefore, it has good predictive performance for time series [44]. In the method proposed in this paper, the LSTM network is used to predict the various VMD modes $\{DR_{t,n}^{k}\}=\{DR_{t}^{k},\ldots ,DR_{t+n-1}^{k}\},k=1,\ldots ,K$ and the residual sequence $\left\{DR_{t,n}^{res}\}=\{DR_{t}^{res},\ldots ,DR_{t+n-1}^{res}\right\}$ of the 5G DRTT series. It is crucial to model the LSTM network for the 5G DRTT series before implementing predictions. The flow chart shown in Figure 7 illustrates the three main steps of the LSTM network modeling process:

Step 1. Data preprocessing and input:

Considering that a large sample size will result in a long training time, 2000 DRTT data were selected as the training and prediction dataset for the LSTM network. We used the first 80% of the dataset as the training set and the remaining 20% as the test set to evaluate the predictive performance of the modeled LSTM network. Then, the following min–max normalization method was used to standardize the data to within [0, 1]:(9)$$DR_{t}^{\prime }=\frac{DR_{t}-\min\left(\left\{DR_{t},t\in T\right\}\right)}{\max\left(\left\{DR_{t},t\in T\right\}\right)-\min\left(\left\{DR_{t},t\in T\right\}\right)}$$
in which $\left\{DR_{t},t\in T\right\}$ is the DRTT series composed of all DRTT data, and $\left\{DR_{t}^{\prime },t\in T\right\}$ is the standardized DRTT series, T∈Z, represents the time range of the chosen DRTT series.

Step 2. LSTM parameter settings:

In this study, the LSTM network used to predict standardized DRTT values was defined as a four-layer network consisting of an input layer, a fully connected layer, and two hidden layers. Each hidden layer contained 50 neurons. The other hyperparameters that need to be set include Epoch, Batch Size, and Optimizer. The values of these hyperparameters can be determined using complex hyperparameter optimization methods, such as grid search. In this study, the specific values of the aforementioned hyperparameters were determined through sensitivity analysis of the model, as detailed in Section 7.2.

For LSTM networks, except for the above hyperparameters, there were two important hyperparameters, which were the input size *m* (also known as the time step) and the output size *n*. After the above hyperparameters were determined, other parameters in the LSTM network could be identified. Subsequently, *m* known DRTT values could be used to predict *n* unknown DRTT data. In the field of deep learning, the selection of *m* and *n*, like other hyperparameters, requires a lot of experimentation, such as using grid search [56] to find the appropriate values. During this search process, once the set of hyperparameters is changed, we have to retrain the LSTM network and analyze whether its prediction results meet the requirements. Therefore, common deep learning parameter tuning methods are time-consuming and complex. To this end, we propose a method for determining the input and output sizes of LSTM networks based on time series analysis results.

In the tuning process of VMD parameters in Section 6.1, we conducted correlation analysis on this *K* + 1 subsequence. The analysis indicated that m−n=81. Therefore, we set the input and output sizes of these *K* + 1 LSTM networks to *m* = 82 and *n* = 1.

Step 3. Modeling *K* + 1 LSTM networks:

LSTM network modeling is a complex process of parameter identification. The basic unit of the LSTM network is shown in Figure 8.

In this paper, the gradient descent method Adam was used to iteratively update the weight coefficient matrices $W_{f}$, $W_{i}$, $W_{C}$, and $W_{o}$, as well as the bias matrices $b_{f}$, $b_{i}$, $b_{C}$, and $b_{o}$. These parameters were used to update the output value $h_{t}$ of the unit. From the third step in Figure 7, it can be seen that in the proposed VMD-LSTM modeling method the parameters of each LSTM network corresponding to *K* VMD subsequences $\left\{DR_{t,-m}^{1}\},\ldots ,\{DR_{t,-m}^{K}\right\}$ and residual sequence $\left\{DR_{t,-m}^{res}\right\}$ (i.e., a total of *K* + 1 subsequences) are iteratively updated. Compared with the method in [28] that set the stopping condition as the sum of the output values of *K* + 1 sequences meeting the fitting accuracy requirement, our training method for network parameters is to stop fitting the current subsequence only when the epoch, the number of iterations, reaches the set maximum number of iterations. After the iteration calculation of a single subsequence is completed, the epoch value that meets the accuracy requirements of the current subsequence is determined by backtracking the historical information during the network training process of this subsequence. In the former method, as all subsequences use the same number of iterations, overfitting can easily occur, thereby affecting prediction accuracy. As a comparison, our method will adaptively select the number of iterations for each of the *K* + 1 LSTM networks based on the changes in their respective loss functions during the iteration process to avoid overfitting.

When the optimal epochs are determined, their corresponding weight coefficient matrix and bias matrix are also calculated accordingly. During the process of modeling the LSTM network, the output value of the LSTM unit is updated and calculated according to the following formulas:(10)$$f_{t}=\sigma \left(W_{f}\left[h_{t-1},x_{t}\right]+b_{f}\right)$$
(11)$$i_{t}=\sigma \left(W_{i}\left[h_{t-1},x_{t}\right]+b_{i}\right)$$
(12)$${\tilde{C}}_{t}=\tanh\left(W_{C}\left[h_{t-1},x_{t}\right]+b_{C}\right)$$
(13)$$C_{t}=f_{t}\times C_{t-1}+i_{t}\times {\tilde{C}}_{t}$$
(14)$$o_{t}=\sigma \left(W_{o}\left[h_{t-1},x_{t}\right]+b_{o}\right)$$
(15)$$h_{t}=o_{t}\times \tanh\left(C_{t}\right)$$

At this point, the LSTM network has been modeled. According to the prediction method for a single sequence in Figure 4, the predicted sequences $\left\{D{\hat{R}}_{t,n}^{1}\},\ldots ,\{D{\hat{R}}_{t,n}^{K}\right\}$ and the predicted residual sequence $\left\{D{\hat{R}}_{t,n}^{res}\right\}$ can be obtained using (10)–(15) based on their respective weight coefficients and biases. By using (2) to predict the DRTT series and finally performing inverse difference according to (4), the predicted sequence $\left\{{\hat{R}}_{t},\ldots ,{\hat{R}}_{t+n-1}\right\}$ of the 5G RTT series can be obtained.

## 7. Results and Discussion

We used the machine learning framework TensorFlow-GPU and Keras package to construct the LSTM network. The code was written in Python 3.7 and executed on a workstation with an NVIDIA Quadro P2000 GPU and an Intel Xeon E5-2650 v4 CPU.

### 7.1. Prediction Performance Evaluation Metrics

We use a dual AGV synchronous control system based on 5G as an example to introduce the metrics for evaluating the performance of the proposed VMD-LSTM model in predicting 5G RTT. In this system, the central controller sends commands to the two AGVs via the 5G network according to a fixed synchronous control cycle, enabling them to clarify their respective motion modes within the current cycle, thereby ensuring the accuracy of synchronous movement. In this system, 5G RTT refers to the time delay between when the AGV receives the command and when the controller sends that command, which has the following impacts on the accuracy and stability of the system:(1)Before sending motion commands, the central controller must take into account the offsets caused by the latency time on the AGV’s position and speed in order to calculate the appropriate command values. Therefore, the accuracy of RTT prediction determines the accuracy of the synchronous motion commands;(2)When the 5G RTT exceeds the AGV’s synchronous control cycle value, it indicates that the commands issued by the central controller were not received by the AGV within the current cycle. If the AGV misses multiple synchronous commands, the system may lose stability. To avoid this situation, a compensation control strategy should be preset and then activated when it is predicted that the AGV will soon be unable to receive synchronous commands. Therefore, accurately predicting whether control commands can be received within each control cycle is crucial for the stability of the system.

Therefore, this section categorizes the evaluation metrics for the model’s predictive capability into two types: RTT prediction accuracy metrics and early warning accuracy metrics for control instability issues.

#### 7.1.1. Prediction Accuracy Metrics

The evaluation indicators for the prediction accuracy of time series generally include root mean square error, i.e., *RMSE*, and mean absolute percentage error, i.e., *MAPE*. The formulas for their calculation are given below:(16)$$RMSE=\sqrt{\frac{1}{T_{N}}\sum_{t=1}^{T_{N}}{\left({\hat{R}}_{t}-R_{t}\right)}^{2}}$$
(17)$$MAPE=\frac{1}{T_{N}}\sum_{t=1}^{T_{N}}\left|\frac{{\hat{R}}_{t}-R_{t}}{R_{t}}\right|\cdot 100\%$$
where $\left\{R_{1},R_{2},\ldots ,R_{T_{N}}\right\}$ denotes the original time series, and $\left\{{\hat{R}}_{1},{\hat{R}}_{2},\ldots ,{\hat{R}}_{T_{N}}\right\}$ denotes the predicted time series.

#### 7.1.2. Early Warning Accuracy Metrics for Control Issues

As shown in Equation (18), we propose a metric *TC*. Here, $TO_{t}$ is used to characterize the early warning accuracy of the RTT prediction method for the *t*th control cycle; when this value is 1, it indicates a successful prediction of whether the controlled object can or cannot receive the control command within that cycle. Therefore, *TC* can be used to represent the early warning accuracy of the RTT prediction method for the entire motion control process (all control cycles); the closer this value is to 100%, the stronger the early warning capability of the RTT prediction method regarding control instability issues.
$$TC=\frac{\sum_{t=1}^{T_{N}}TO_{t}}{T_{N}}\cdot 100\%$$
(18)$$TO_{t}=\left\{\begin{array}{l} 1,({\hat{R}}_{t}-\Delta T)\cdot (R_{t}-\Delta T)>0\\ 1,{\hat{R}}_{t}=R_{t}=\Delta T\\ 0,({\hat{R}}_{t}-\Delta T)\cdot (R_{t}-\Delta T)<0\\ 0,else \end{array}\right.$$

The early warning metric *TC* can be further used for the following functions:

(1)Indicating when to retrain the model.

In certain scenarios, users set the control cycle as a fixed constant. When the *TC* value of our VMD-LSTM model indicates that its early warning capability for the control system is no longer satisfactory, it is necessary to retrain the model to help maintain the stability of the control system.

(2)Indicating the setting of the control cycle.

In some cases, the control cycle can be chosen within a certain range. In such cases, *TC* values can be calculated based on different control cycles, and the specific value of the control cycle can be guided by the desired *TC* value.

### 7.2. Impact of Hyperparameters on Predictive Performance

In order to analyze the impact of different hyperparameter settings on the predictive performance of the model, we trained the model 10 times with the same set of hyperparameters. The median of the 10 different test error *RMSE* values obtained is used to represent the predictive performance of the model under that hyperparameter setting.

#### 7.2.1. Epoch

We set the number of subsequences (*K*) to 7 as previously mentioned. After VMD decomposition, we obtained 7 subsequences and 1 residual sequence, making a total of 8 sequences. For simplicity, we will refer to all of them as subsequences hereafter. Then, we calculated the difference between test error and training error $\delta_{s}$ for 8 subsequences under various epoch values. As previously mentioned, through 10 repeated training and testing, we obtained the median error $\delta_{s}^{mid}$ of $\delta_{s}$ for each parameter combination, which is shown in Table 5. In which, numbers enclosed in * indicate the minimum data value in each column, representing the best performance achieved by the subsequence. Therefore, the epoch value corresponding to this minimum $\delta_{s}^{mid}$ value is the most suitable epoch for the subsequence.

As mentioned in Section 6.2, we set a common maximum epoch value for all subsequences. The model then adaptively selects the optimal epoch value for each subsequence individually, ensuring none exceed the set maximum. Based on the analysis of the data in the table above, we set this maximum epoch value to 200.

#### 7.2.2. Batch Size

Figure 9 illustrates the impact of batch size on prediction performance. The red line in each box represents the median of the *RMSE*, indicating the average error in RTT prediction performance. The overall range of each box represents the variability in the model’s prediction *RMSE*. A smaller average error and lower variability indicate good and repeatable prediction performance. The figure contains ten boxes, showing the prediction error results of the VMD-LSTM model when the batch size values are even numbers ranging from 2 to 20. It can be observed that considering the average level and variability of prediction error, the optimal batch sizes for comprehensive performance are 6, 8, and 16. However, the smaller the batch size, the longer the time required to train the model. Therefore, considering the model training time, we set the batch size to 16 when training the VMD-LSTM model.

#### 7.2.3. Optimizer

Figure 10 compares the prediction performance of three optimizers. The results show that the variability of the RMSprop algorithm is significantly greater than that of the other two optimizers, while SGD has the largest prediction error among the three. Therefore, Adam is more suitable for predicting the 5G RTT data measured in the factory setting in this study.

#### 7.2.4. Time Step

The time step of LSTM, which is the value of the length *m* of a single input DRTT series in the LSTM prediction process, will affect the training effect of LSTM. As described in Section 6, in our VMD-LSTM prediction method, we used the correlation analysis method to solve for the correlation coefficient of DRTT. The results indicated that the lag of our sample DRTT was 82, so we set the time step of LSTM to 82.

We evaluated the performance of LSTM prediction with a time step of around 82 and at other values, and we present the performance indicator values in Table 6, where ‘*’ represents the time step used in this study. It can be seen that the DRTT sample sequence used for training and testing network performance has higher predictability when the time step value is the partial autocorrelation truncation order of the sequence. The results of the time steps at 10–70 indicate that the prediction performance of the sequence will gradually deteriorate as the time step value deviates from the partial autocorrelation truncation order. This means that there is a certain degree of negative correlation between the prediction performance and the degree of time step deviation. Meanwhile, when the time step is around 82, the predictive performance of the sequence fluctuated within a small range because the truncation order of subsequences also fluctuated around 82. As a further refinement of our prediction method, the corresponding LSTM time step value can be selected based on the partial autocorrelation truncation properties of each subsequence. However, because this further exacerbates the complexity of the prediction method and the partial autocorrelation truncation features of each subsequence after VMD are basically consistent, we used the same time step for each subsequence.

#### 7.2.5. *K*

Although we optimized the *K* value to 7 based on the characteristics of the subsequences during VMD modeling, it is still necessary to analyze the impact of *K* on the overall VMD-LSTM model. As shown in Figure 11, the median of the test error *RMSE* decreases as the *K* value increases. Larger *K* values significantly improve the model’s performance, but the decrease in test error slows down when the *K* value reaches 20. By comparing this relationship curve with the impact of other hyperparameters on the test performance mentioned earlier, it can be observed that the sensitivity of the proposed model’s performance to the *K* value is significantly higher than that of other parameters.

However, a larger *K* value means more models need to be trained, which results in greater computational time. To avoid spending excessive time training models, we conducted a further analysis of the *K* value from the perspective of model transferability.

As described in Table 3, there are a total of eight datasets. We used 2000 samples from Dataset 1 (16 ms 32 byte) for VMD-LSTM modeling. For each fixed *K* value, the sequence composed of the data is decomposed into *K* + 1 subsequences. We train a separate LSTM sub-model for each of these *K* + 1 subsequences, resulting in *K* + 1 sub-models, referred to as a model set. Since *K* can take integer values from 1 to 30, we trained a total of 30 model sets, with the *K*th model set consisting of *K* + 1 sub-models.

Next, we used four datasets as the transferability test datasets, specifically Dataset 1, Dataset 4, Dataset 5, and Dataset 8. Taking Dataset 8 as an example to illustrate the testing process.

(1)Randomly extract a continuous 2000 data points from Dataset 8 to test the prediction errors of the existing models.(2)For each fixed *K* value, the trained *K*-th model set is used to predict the data selected in step (1), and the corresponding RTT prediction error *RMSE* is calculated. This results in 30 *RMSE* values for different *K* values (from 1 to 30), referred to as a set of *RMSE* values for Dataset 8.(3)Steps (1) and (2) are repeated a total of 10 times, resulting in 10 sets of *RMSE* values.

After the above three steps, each fixed *K* value has 10 $RMSE_{K}$ values in the 10 sets of *RMSE* values, and the median of these values is calculated as ${RMSE}_{K}^{mid}$.

The same testing process was applied to the other three test datasets. Figure 12 shows the relationship between ${RMSE}^{mid}$ and *K* values for different datasets. It can be observed that The ${RMSE}^{mid}$ for all four datasets shows a decreasing trend with fluctuations as *K* increases. However, when *K* is less than 7 or greater than 13, the ${RMSE}^{mid}$-*K* curves exhibit significant fluctuations, with poor consistency between the datasets. In contrast, when *K* is between 7 and 13, ${RMSE}^{mid}$ shows a stable decreasing trend, and the curves of the four datasets demonstrate good consistency. Therefore, setting *K* in this range can better accommodate changes in 5G RTT characteristics due to variations in the factory environment, enhancing transferability across different datasets. Considering that a larger *K* value leads to longer training and prediction times, and that there is not much improvement when *K* is 13 compared to *K* is 12, so we set *K* to 12.

### 7.3. Comparison of Prediction Performance of Different Methods

We propose a time series analysis-based VMD-LSTM prediction method, which takes into account the non-stationarity of the 5G RTT series and takes the first-order differential sequence of the original sequence as the input of the VMD-LSTM model. At the same time, our calculation and analysis of the partial autocorrelation coefficients of the time series were also used to optimize the parameters of VMD and the time step value of LSTM. Section 7.2 validated the usefulness of our method of determining LSTM time step values through correlation coefficients. In this section, we use several different prediction methods for RTT series as benchmarks for the proposed method and compare their prediction performance. These methods are as follows:

a.LSTM

This method directly takes 5G RTT values as inputs to the LSTM network and predicts future RTT values.

b.EEMD-LSTM

EEMD is an optimization method for the signal decomposition method EMD, which performs better on nonlinear and non-stationary signals than EMD. Therefore, EEMD-LSTM networks are also commonly used for time series prediction [25].

c.VMD-LSTM

The VMD-LSTM method is the foundation of the VMD-LSTM prediction method based on the time series analysis proposed in this paper. Kaihan Wu, Junhui Lu, et al. demonstrated the excellent performance of the VMD-LSTM method in predicting network traffic [29]. Therefore, it is necessary to compare the prediction performance of our method with the VMD-LSTM method.

d.The time series analysis-based VMD-LSTM prediction method proposed in this paper.

The indicators of the predictive performance of each method are listed in Table 7. Among them, the accuracy of directly predicting the original sequence using LSTM is the worst. This is because the 5G RTT series is unstable and the components of the random sequence have a significant impact. Therefore, LSTM and other methods that directly predict the sequence will have relatively poor results. This is exactly the original intention of proposing our time series analysis-based VMD-LSTM prediction method.

On the contrary, the other three methods first preprocess the original sequence to improve its predictability by reducing its complexity. As shown in Table 7, the prediction accuracy of these three methods is superior to that of the individual LSTM prediction method. Among them, EMD and VMD are used as signal decomposition methods in the EMD-LSTM method and VMD-LSTM method, respectively. However, as mentioned earlier, due to the involvement of three important hyperparameters in the VMD method, their selection is usually complex. Unless they are carefully tuned, it will result in a smaller improvement in the prediction performance of the VMD-LSTM method on complex time series compared to using the LSTM method alone. The use of the EEMD method is relatively simple, without the need for fine-tuning of hyperparameters.

Finally, we would like to focus on analyzing the method we propose. From the results in Table 7, it can be seen that our method has significantly improved prediction accuracy compared to other methods. This is because we propose a simple method for tuning VMD parameters, which makes the results of VMD reliable, providing a guarantee for improving the prediction accuracy of the entire method. On the other hand, due to the first-order differencing of the 5G RTT series in our method, we obtained the DRTT series with stationarity, which makes the prediction accuracy of LSTM for the first-order differencing sequences much higher than that for the original sequence. Subsequently, a simple reconstruction is required to achieve high-precision prediction of the original 5G RTT series. See Table 7.

Due to the poor prediction performance of other methods on the RTT series, only the prediction performance of the method we proposed is shown in Figure 13. As shown in the figure, for complex RTT series, although our method’s predicted values have a certain deviation from the actual values in terms of amplitude, the predicted results show good follow-up; that is, there is little time lag. Meanwhile, due to the RTT data of the test set being sent out with an interval of 16 ms, according to (18), the value of *TC* is 98.5%, which proves that our proposed time series analysis-based VMD-LSTM prediction method is helpful for the application of 5G in the industrial field.

## 8. Conclusions

In order to apply 5G technology to factory environments with low latency requirements for data transmission, we proposed a 5G RTT time series analysis-based VMD-LSTM prediction method. To obtain the performance data of 5G technology applied in real factory scenarios, we designed an on-site testbed from SPLC contracting (pinging a packet) to MPLC and recorded 5G RTT data. All the devices in the test network were existing commercial products rather than prototypes, and the RTT measurement method is also relatively simple. The statistical analysis of the 5G RTT data indicated that their average value was around 11 ms, which is better than the WIA-FA standard [57] and can better meet the low latency requirements of industrial data transmission. The ADF test method was used to prove that the 5G RTT time series is non-stationary. The DRTT series, after the first-order difference, was transformed into a stationary sequence. Combining the autocorrelation and partial autocorrelation coefficients of the sequence, we proposed a VMD-LSTM prediction method based on time series analysis. Compared with other prediction methods that combine EEMD, VMD, and LSTM, this method has the best prediction accuracy.

The time series analysis-based VMD-LSTM prediction method proposed in this paper can be used to evaluate network performance based on business requirements. In the 5G network, by establishing a corresponding VMD-LSTM model for RTT prediction and comparing the actual RTT with the predicted one, abnormal values can be identified, and their causes can be analyzed, guiding the improvement of the 5G network. Furthermore, we have defined an indicator *TC* related to the control cycle to characterize whether the prediction method can accurately predict whether the RTT value exceeds the control cycle. In the field of industrial control, this method can be used for RTT prediction and motion control compensation in the sampling period of control systems, reducing the impact of control instruction packet loss caused by RTT, as well as improving the robustness of the control algorithms. In addition, *TC* can also be used to indicate when to retrain the model and to indicate the setting of the control cycle.

The work in this paper is based on the real experimental data of the 5G-R15 version, and the proposed method can also be used for higher versions of 5G networks. To achieve real-time prediction, single multi-step prediction can be carried out in the future. Considering that the accuracy of multi-step prediction usually decreases with an increase in the prediction step size, indicators such as signal-to-noise ratio, load, and RTT can be used as inputs for multi-factor LSTM prediction, thus balancing prediction accuracy and real-time prediction performance.

## Figures and Tables

**Figure 1 sensors-24-06542-f001:**
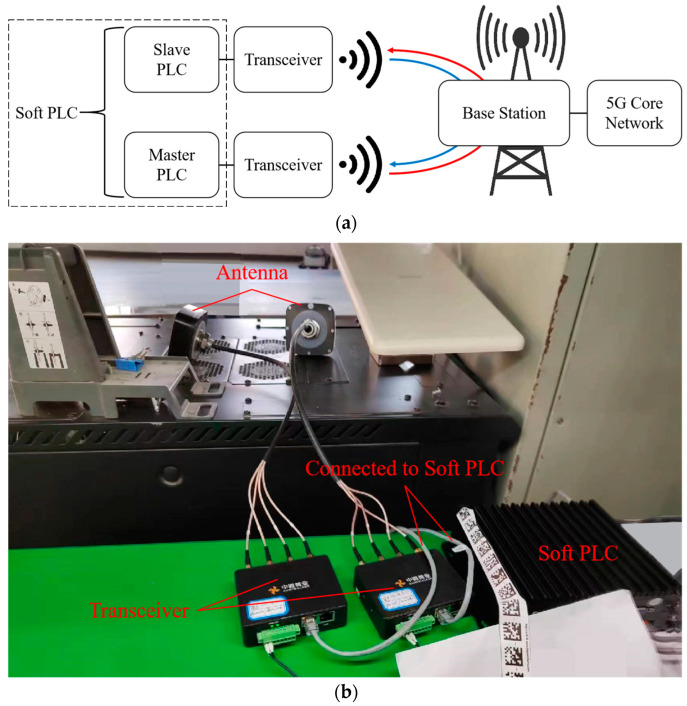
Structure of the test network. (**a**) Schematic architecture. (**b**) Real factory scenario.

**Figure 2 sensors-24-06542-f002:**
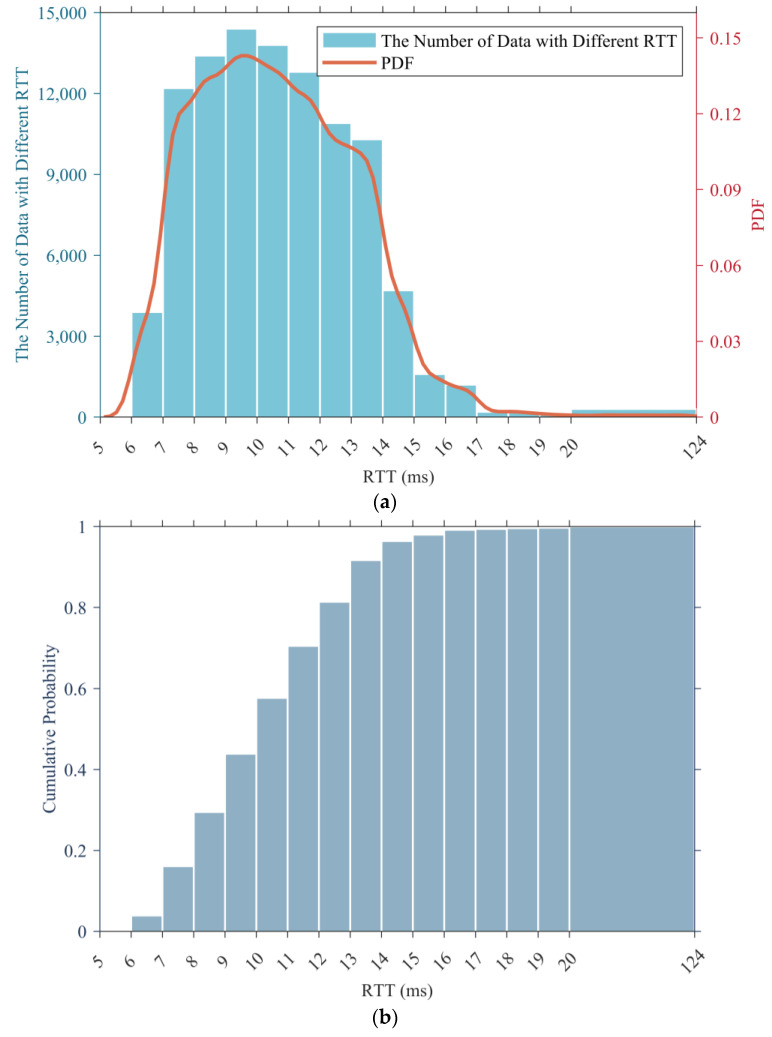
Statistical results for dataset 1. (**a**) Histogram and PDF. (**b**) Cumulative histogram.

**Figure 3 sensors-24-06542-f003:**
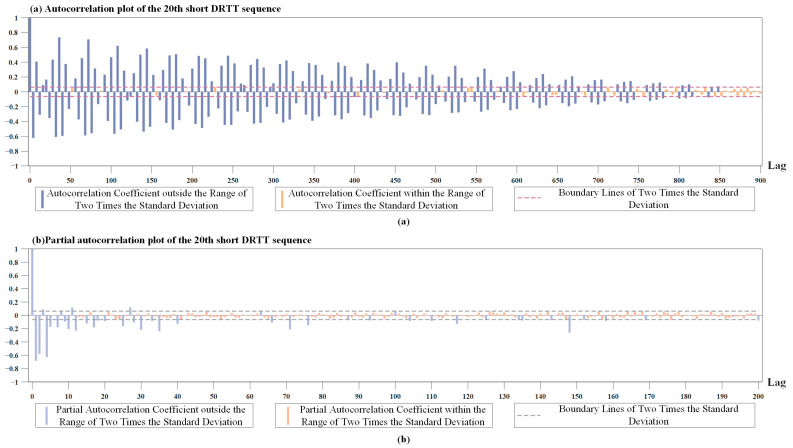
Results of the autocorrelation coefficient and partial autocorrelation coefficient. (**a**) The autocorrelation coefficient of the short DRTT sequence. (**b**) The partial autocorrelation coefficients of the short DRTT sequence.

**Figure 4 sensors-24-06542-f004:**
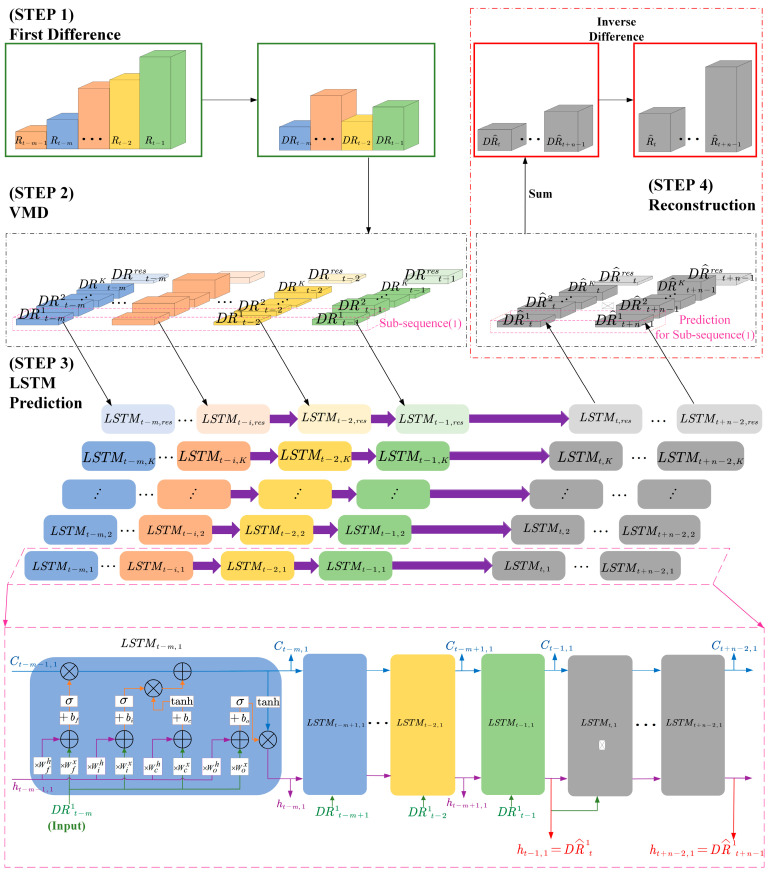
Framework diagram of the time series analysis-based VMD-LSTM prediction method for 5G RTT.

**Figure 5 sensors-24-06542-f005:**
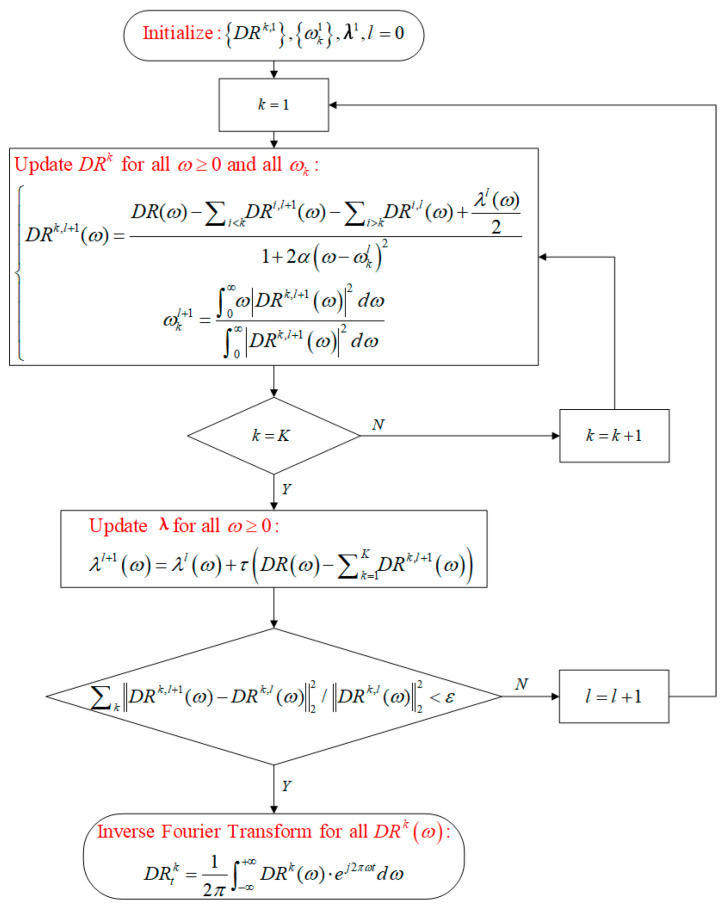
VMD flow chart for 5G DRTT.

**Figure 6 sensors-24-06542-f006:**
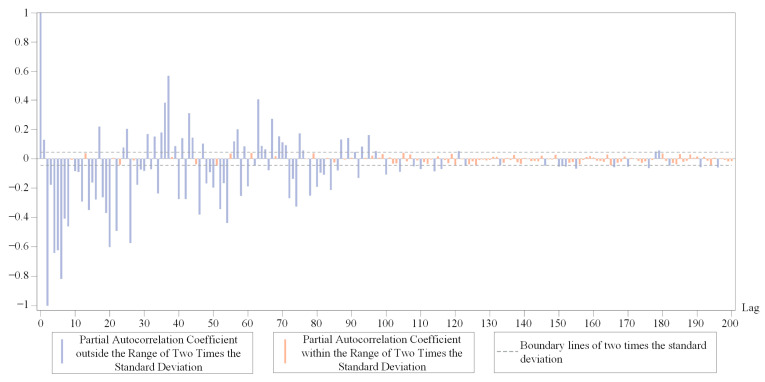
Partial autocorrelation coefficient of a subsequence.

**Figure 7 sensors-24-06542-f007:**
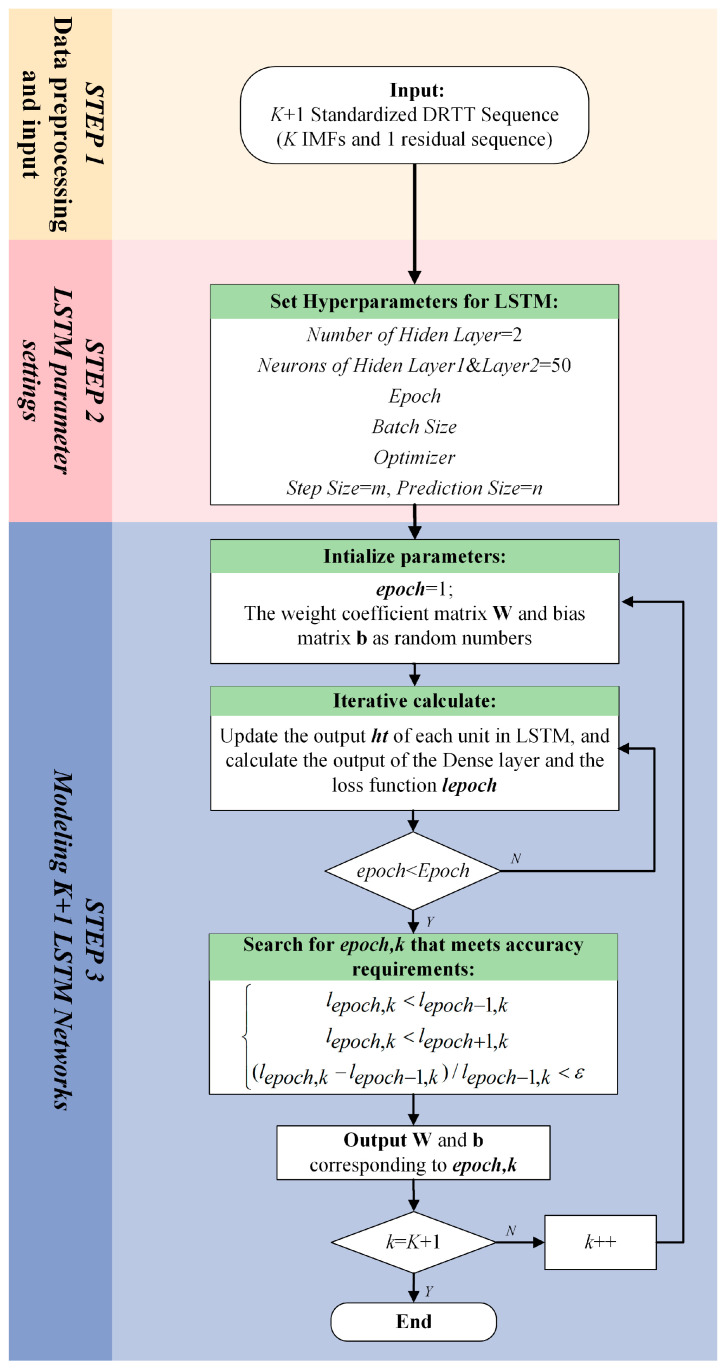
LSTM network modeling process.

**Figure 8 sensors-24-06542-f008:**
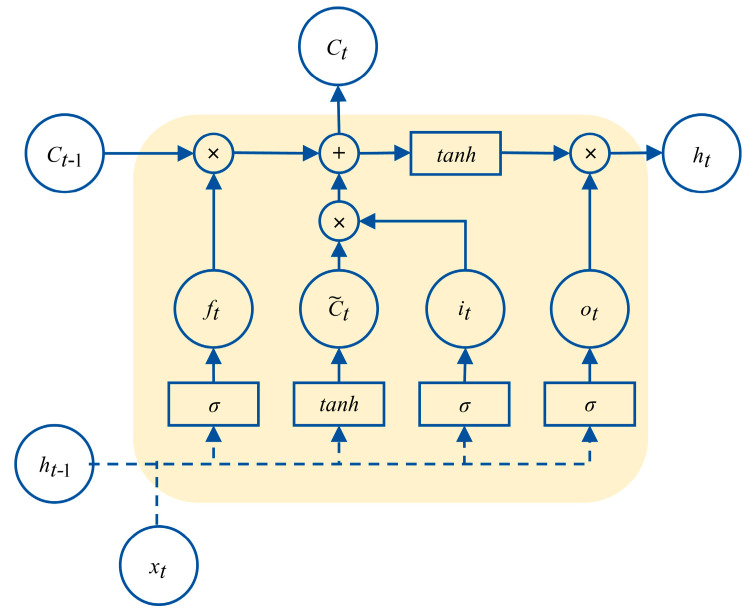
Basic unit of the LSTM network.

**Figure 9 sensors-24-06542-f009:**
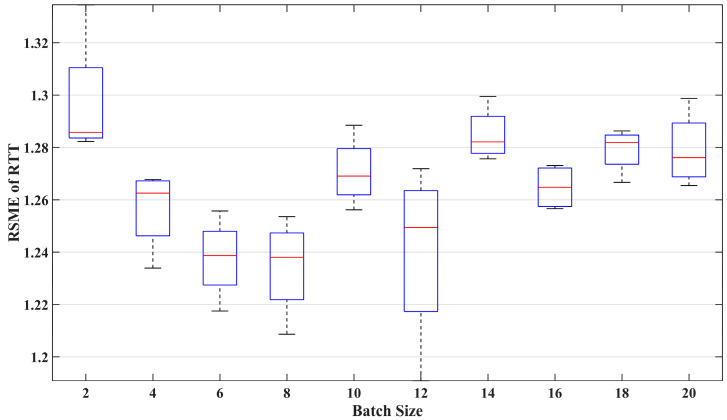
The impact of batch size on prediction performance.

**Figure 10 sensors-24-06542-f010:**
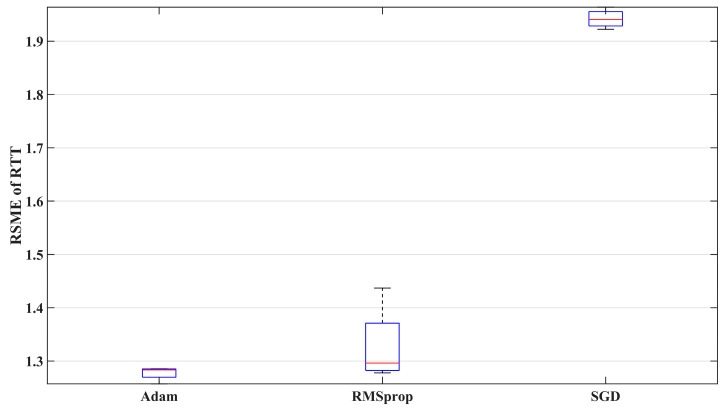
The prediction performance of different optimizers.

**Figure 11 sensors-24-06542-f011:**
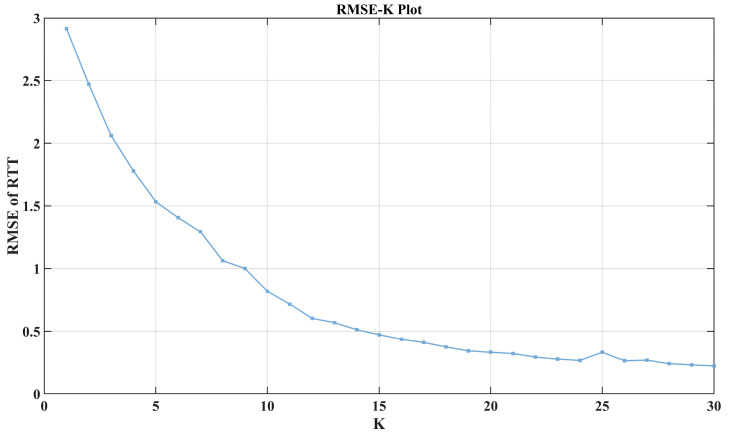
The median of the test error *RMSE* decreases as the *K* value increases.

**Figure 12 sensors-24-06542-f012:**
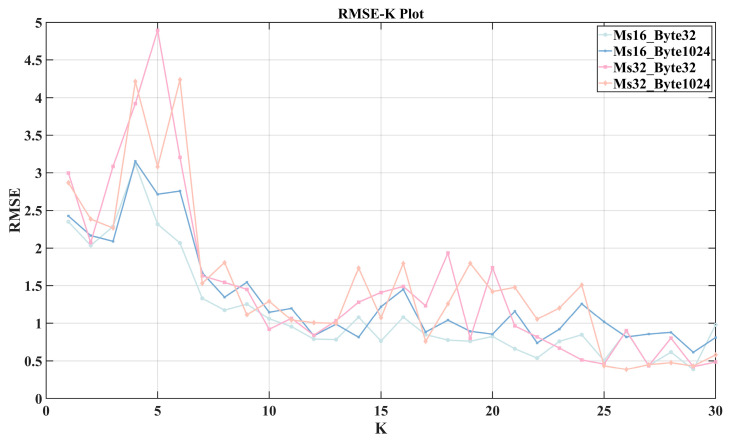
The relationship between ${RMSE}^{mid}$ and *K* values for different datasets.

**Figure 13 sensors-24-06542-f013:**
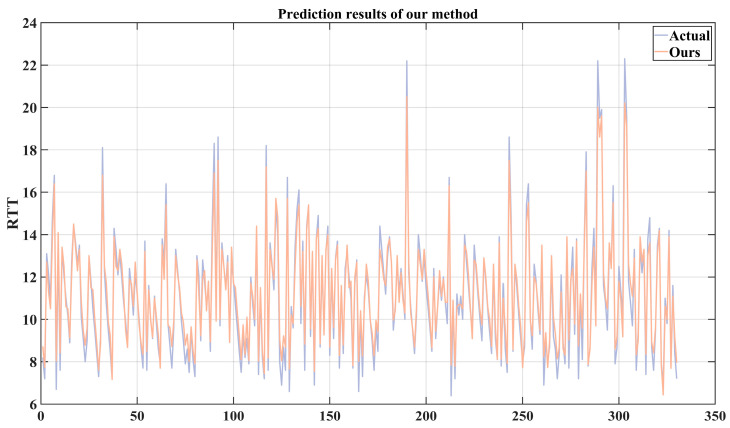
The predictive performance of the proposed method.

**Table 1 sensors-24-06542-t001:** MCS index table.

MCS Index	Modulation Index	Target Code Rate R × [1024]	Spectral Efficiency
0	2	120	0.2344
1	2	157	0.3066
2	2	193	0.3770
3	2	251	0.4902
4	2	308	0.6016
5	2	379	0.7402
6	2	449	0.8770
7	2	526	1.0273
8	2	602	1.1758
9	2	679	1.3262
10	4	340	1.3281
11	4	378	1.4766
12	4	434	1.6953
13	4	490	1.9141
14	4	553	2.1602
15	4	616	2.4063
16	4	658	2.5703
17	6	438	2.5664
18	6	466	2.7305
19	6	517	3.0293
20	6	567	3.3223
21	6	616	3.6094
22	6	666	3.9023
23	6	719	4.2129
24	6	772	4.5234
25	6	822	4.8164
26	6	873	5.1152
27	6	910	5.3320
28	6	948	5.5547
29	2	Reserved	
30	4	Reserved	
31	6	Reserved	

**Table 2 sensors-24-06542-t002:** Parameters of the 5G test system.

Parameters	Value
Sub-frame Ratio	7:3
Carrier Frequency	3.5 GHz
Bandwidth	100 MHz
Cyclic Prefix (CP) Length	Normal
Intercarrier Spacing	30 kHz
Channel Coding	Service Channel: LDPC
	Control Channel: Polar
Modulation Scheme	AMC
Frame Duration	10 ms
Number of Slots	1 Frame: 20 Slots
Number of Symbols	1 Slot: 14 Symbols
Duplex Mode	Time Division Duplex (TDD)
NR Band	3.5 GHz–3.6 GHz

**Table 3 sensors-24-06542-t003:** Packet settings and test results of 5G RTT.

Dataset Number	Packet Sending Interval (ms)	Packet Length (byte)	Min RTT (ms)	Max RTT (ms)	Mean (ms)	Standard Deviation (ms)	Skewness	Kurtosis
1	16	32	6.4	123.7	10.57	2.61	8.60	352.65
2	16	128	6.3	39.5	10.75	2.66	1.30	5.32
3	16	640	6.4	29.3	11.16	3.03	0.96	1.97
4	16	1024	6.5	34.4	11.13	2.74	0.57	0.56
5	32	32	6.5	186.9	11.16	3.45	19.55	902.73
6	32	128	6.4	186.0	11.31	3.86	18.11	694.31
7	32	640	6.4	99.2	11.10	2.67	4.24	120.96
8	32	1024	6.5	161.5	11.23	3.32	15.00	584.45

**Table 4 sensors-24-06542-t004:** Augmented Dickey–Fuller Tests of Short 5G RTT Sequences.

Short Sequence Number	Lag	*p*-Value	τ-Statistic	Critical Value (0.05)
1	27	0.6718	0.0548	−1.9413
11	66	0.9374	1.1642	−1.9413
21	36	0.8666	0.7028	−1.9413
31	29	0.7129	0.1671	−1.9413
41	14	0.8302	0.5309	−1.9413

**Table 5 sensors-24-06542-t005:** $\delta_{s}^{mid}$ for subsequences under various epoch values.

	Subsquence Index	1	2	3	4	5	6	7	8
Epoch									
**25**		***0.0030***	0.0050	0.0040	0.0070	0.0040	0.0060	0.0025	***0.1930***
**50**		0.0030	***0.0045***	0.0040	***0.0065***	0.0075	***0.0055***	0.0020	0.3535
**100**		0.0030	0.0065	0.0040	0.0075	***0.0040***	0.0065	0.0020	1.1585
**200**		0.0035	0.0125	***0.0020***	0.0085	0.0045	0.0075	***0.0010***	1.4915
**300**		0.0040	0.0205	0.0110	0.0100	0.0085	0.0080	0.0030	1.1625
**400**		0.0050	0.0380	0.0135	0.0175	0.0090	0.0080	0.0030	1.4560

Note: Numbers enclosed in * and * indicate the minimum data value in each column.

**Table 6 sensors-24-06542-t006:** Comparison of prediction performance under different time steps.

Time Step	*RMSE* (ms)	*MAPE* (%)
80	1.06	7.682
81	1.04	7.652
82 *	1.05	7.655
83	1.05	7.701
84	1.06	7.654
70	1.069	7.925
50	1.08	8.288
30	1.112	8.657
10	1.134	8.926

Note: Symbol * represents the value of the time step used in this study.

**Table 7 sensors-24-06542-t007:** Comparison of prediction performance of different methods.

Methods	*RMSE* (ms)	*MAPE* (%)
LSTM	2.891	20.900
EEMD-LSTM	2.451	16.177
VMD-LSTM	2.786	19.132
Ours	0.608	4.481

## Data Availability

The raw data supporting the conclusions of this article will be made available by the authors upon request.

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
