# Peer review of "Performance Analysis and Prediction of 5G Round-Trip Time Based on the VMD-LSTM Method"

_sensors, 2024, doi:10.3390/s24206542_

Round 1

Reviewer 1 Report (New Reviewer)

Comments and Suggestions for Authors

Main comments

This article is very lengthy and could easily be shortened and sharpened a bit without warranting the main content of the paper. Extensive related literature is included in the bibliography. The novelty is limited as the paper focuses more on performance studies of existing technologies. As such, the technical contribution is reasonable. However, in addition to the length of the paper, there is plenty of room for improvement in the layout of the paper. Minor comments below list examples of problematic issues and how to solve them.      

Minor comments

The next comments are more detailed comments/remarks (many of which editorial/general outlook in nature) that should be taken into consideration in the revision.

-          Page 3, Introduction, line number 108 (and onwards): Separate abbreviated notations ‘i.e.’ and ‘e.g.’ with commas on both sides throughout the manuscript while they appear in the middle of the sentence and in this case put the comma after e.g.  

-          Page 7, Section 3, Table 2, 3rd row under Value column: M -> MHz

-          Page 10, Section 5 onwards: Ensure that mathematical symbols appear systematically in the same format throughout the manuscript both in the equations and in the text (italics, subscripts, superscripts, font sizes, spacing, punctuation, etc.).

-          Page 10, Section 5, line number 296: TN -> and TN

-          Page 10, Section 5.1, line number 304: Avoid beginning a sentence with a small letter of small-lettered symbol. It is easy to get rid of this issue by adding something in front, e.g., Symbol …).

-          Page 10, Section 5.1, line number 312: The expression beginning as ‘According to …’ should be revised for better readability.

-          Page 10, Section 5.1, line number 318: Ambiguity in the expression could be avoided by stating, e.g., ‘None of the short sequences shown in Table 4 satisfied …’.

-          Page 10, Section 5.1, line number 326: Put a full-stop at the end of the equation as it closes the phrase. Similarly, embed the rest of the equations as parts of the preceding and following text with proper punctuation and indentation. To refer to a particular equation it is adequate to use the number in parentheses as such without ‘Formula’ in front unless the reference opens a sentence.

-          Page 11, Section 5.2, line numbers 344 and 349: I would not use the word ‘order’ in this context (lag) as it can cause confusion. The same issue appears again in Section 6.1.

-          Page 13, Section 6, Figure 4: This figure includes too small and blurry details to be readable. Larger font size and better resolution is needed. Later on, Figs. 6 and 9 have similar problems.

-          Page 14, Section 6, line number 417: subsequences respectively. -> subsequences, respectively.

-          Page 18, Section 6.2, line number 554: choosed -> chosen   

Comments on the Quality of English Language

The language is mostly fluent and readable. However, some language-related revisions are recommended in minor comments.

Author Response

Reviewer 2 Report (New Reviewer)

Comments and Suggestions for Authors

The paper aims to analyze and predict the round-trip time (RTT) of 5G communication in industrial environments, particularly those that require low latency (e.g., smart factories). The paper proposes a novel method combining Variational Mode Decomposition (VMD) and Long Short-Term Memory (LSTM) networks to predict 5G RTT.

The paper is interesting but can be intensively improved. Please consider my comments below:

  • Since paper is oriented more towards 5g, authors should also discuss potential to future 6g systems [R1,R2], which must be included in intro. Also, there are related 5G RTT works that are missing such as [R3,R4].
  • The conducted state of the art is incomplete. Comprehensive comparisons of the state-of-the-art works should be provided in a balanced and well structured way.
  • The paper introduces some custom evaluation metrics, such as TC related to the control cycle, but does not provide a clear rationale for their use or explain how these metrics relate to more commonly used measures in the field. This makes it difficult for readers to measure the significance of the results.
  • The description of the experimental setup, including hardware and software configurations, is insufficient. The lack of details such as network conditions, environmental factors, and hardware specifications makes it challenging to replicate the study or understand the conditions under which the results were obtained.
  • The paper focuses on mathematical formulations and technological specifics, with insufficient attention paid to the practical implications and difficulties that can arise when applying the suggested approach in actual industrial settings. Please explain the different equations appearing such as (5) and (6).
  • Why is there red colored fonts ? Why are there strikethrough text ?
  • VMD combined with an LSTM network for predicting the 5G RTT is employed. However, the justification for selecting this specific combination is not explained. While the VMD is used to handle non-stationary time series, the paper does not justify why LSTM was chosen over other neural network architectures, such as Gated Recurrent Units (GRU) or Temporal Convolutional Networks (TCN), which might offer better performance or lower computational costs. 
  • The paper does not discuss challenges involved in the proposed VMD-LSTM , e.g. frequent retraining, integration with existing systems, and real-time data acquisition constraints are not addressed.
  • It is difficult to assess how robust the method is to changes in these parameters or how to select them optimally in different scenarios without a proper sensitivity analysis.
  • Did authors use data that exhibit heavy tails, skewness, or other non-Gaussian characteristics, which could affect the model's performance ?

References 

[R1] A. Bazzi and M. Chafii, “On Outage-Based Beamforming Design for Dual-Functional Radar-Communication 6G Systems,” in IEEE Transactions on Wireless Communications, vol. 22, no. 8, pp. 5598-5612, Aug. 2023, doi: 10.1109/TWC.2023.3235617.

[R2] M. Chafii, L. Bariah, S. Muhaidat and M. Debbah, "Twelve Scientific Challenges for 6G: Rethinking the Foundations of Communications Theory," in IEEE Communications Surveys & Tutorials, vol. 25, no. 2, pp. 868-904, Secondquarter 2023, doi: 10.1109/COMST.2023.3243918.

[R3] del Peral-Rosado, José A., et al. "Exploitation of 3D city maps for hybrid 5G RTT and GNSS positioning simulations." ICASSP 2020-2020 IEEE International Conference on Acoustics, Speech and Signal Processing (ICASSP). IEEE, 2020.

[R4] Zinno, Stefania, et al. "Prediction of RTT Through Radio-Layer Parameters in 4G/5G Dual-Connectivity Mobile Networks." 2023 IEEE Symposium on Computers and Communications (ISCC). IEEE, 2023.

Comments on the Quality of English Language

Can be improved

Author Response

Reviewer 3 Report (New Reviewer)

Comments and Suggestions for Authors

The article analyzes the performance of 5G technology in industrial applications, specifically in reducing round-trip time (RTT) for data transmission. Once the 5G RTT data are found to be non-stationary and aperiodic, so a time series analysis-based variational mode decomposition–long short-term memory (VMD-LSTM) method is proposed for prediction. The results are promising but limited to predictive, with an error of about 7.655%.

It is necessary to be clear why the premises of Release 16 and 17 are the same as those of Release 15 once the new Network Functions are introduced.

It is necessary to keep clear why the article does not consider the specific performance of 5G in reducing round-trip time (RTT) for data transmission in RAN itself. It is necessary to consider the OpenRAN with its xApps and RIC, and Non-RIC should be discussed.

It does not explicitly discuss other sources that could potentially increase RTT. Factors that could potentially increase RTT in any network include Network Congestion, Distance, Network Devices (resources and network hops), Interference, and Quality of Service (QoS).

The abstract has a lot of acronyms that were not defined before its use.

It is unclear what 5GC (5G core) software was used, and it discussed whether it can influence the results.

The specific type of gNodeB (gNB, or 5G base station) used in the study mentioned in the article is not provided in the text. 

It is necessary to explain the testbed components (i.e., hardware and software)

The dataset (artifact) could be available to download and reproduce the research. The citations can be increased, which would help the new researchers.

It is unclear why LightGBM, and Autoregressive Model, for example, were not applied to compare with the VMD-LSTM performance.

Round 2

Reviewer 2 Report (New Reviewer)

Comments and Suggestions for Authors

The authors have addressed my comments.

This manuscript is a resubmission of an earlier submission. The following is a list of the peer review reports and author responses from that submission.

Round 1

Reviewer 1 Report

Comments and Suggestions for Authors

This paper analyzes the performance and prediction of 5G round-trip time based on variational mode decomposition-long 25 short-term memory (VMD-LSTM). In particular, this paper analyzes the statistical performance of the RTT in 5G-R15 communication systems and proposes a VMD-LSTM method to predict the 5G RTT. In general, this paper is written well and easy to follow. However, some major issues need to be carefully addressed.

1. The VMD-LSTM method seems soundness, however, the complexity of the method should be evaluated.

2. The difference between the proposed analysis method and the existing method should be further clarified.

3. Although this paper analyzes the RTT of the 5G communications, the physical layer technique is largely ignored, including the transceiver design, channel coding and modulation. This is related to the reliability and efficiency of the transmission.

4. Especially, the channel coding, especially the LDPC codes, are of great importance to guarantee the transmission reliability of 5G wireless communications.

5. Some state-of-the-art regarding the channel coding and modulation scheme is ignored, such as [A-C].

[A] “Design of Protograph LDPC-Coded MIMO-VLC Systems with Generalized Spatial Modulation," China Communication, 2024, 21(3): 118-136.

[B] “Survey of turbo, LDPC, and polar decoder ASIC implementations,” IEEE Commun. Surveys Tuts., vol. 21, no. 3, pp. 2309–2333, 3rd Quart. 2019.

Comments on the Quality of English Language

This paper analyzes the performance and prediction of 5G round-trip time based on variational mode decomposition-long 25 short-term memory (VMD-LSTM). In particular, this paper analyzes the statistical performance of the RTT in 5G-R15 communication systems and proposes a VMD-LSTM method to predict the 5G RTT. In general, this paper is written well and easy to follow. However, some major issues need to be carefully addressed.

1. The VMD-LSTM method seems soundness, however, the complexity of the method should be evaluated.

2. The difference between the proposed analysis method and the existing method should be further clarified.

3. Although this paper analyzes the RTT of the 5G communications, the physical layer technique is largely ignored, including the transceiver design, channel coding and modulation. This is related to the reliability and efficiency of the transmission.

4. Especially, the channel coding, especially the LDPC codes, are of great importance to guarantee the transmission reliability of 5G wireless communications.

5. Some state-of-the-art regarding the channel coding and modulation scheme is ignored, such as [A-C].

[A] “Design of Protograph LDPC-Coded MIMO-VLC Systems with Generalized Spatial Modulation," China Communication, 2024, 21(3): 118-136.

[B] “Survey of turbo, LDPC, and polar decoder ASIC implementations,” IEEE Commun. Surveys Tuts., vol. 21, no. 3, pp. 2309–2333, 3rd Quart. 2019.

Reviewer 2 Report

Comments and Suggestions for Authors

The manuscript deals with a method to predict delays in 5G devices for industrial applications.

Some general considerations:

1) the manuscript is difficult to read mostly due to unclear, confused, and obscure parts interleaved with trivial descriptions of well-known methods and contradictory statements.

2) Novelty is poor since other papers, not cited by the Authors, have already discussed the same topics using the same, or very similar, methods (see papers [R1]-[R3] shown in the list below).

3) The mathematical part is confused as well, with missing definitions and implicit assumptions while important considerations of the methods are not discussed at all.

4) References to the discussed methods, recalled in Sect. 2, 4, and 5, must be included. Presently, most of them are missing.

5) The proposed method is not well described. I suggest the Authors to use flow charts or pseudo-code fragments to better clarify their proposal.

6) No comparison against other methods from the available literature are presented (see also point #2). 

7) Some important papers are not cited as well: references [R1]-[R3] shows methods very close to the the one proposed by the Authors for delay/RTT estimation of 5G systems. References [R4]-[R6] use a similar VMD-LSTM proposal but adapted for other applications and scenarios, while works [R7]-[R9] discuss some experimental results from real 5G setups.

I suggest the Authors to completely rewrite the manuscript to address these issues.

Some specific considerations:

Abstract - Too generic and with some confusing parts: in the last 20 years, some wireless industrial protocols (e.g. WirelessHART, ISA100.11a, and many others) have been designed with throughput, reliability, and strict latency specifications in mind. This assumption contradicts what is already written in Sect. 1. Please revise.

1. Introduction - This section is somehow confusing and unclear, mostly in the requirement parts related to the industrial wireless networks and their applications. The novelties of the proposal are poor since there is no comparisons against the results of other papers  [R1]-[R3] that exploit methods very close to the one proposed by the Authors. Other papers [R4]-[R6] using very similar methods but in different scenarios may help the Authors to improve their own proposal.

2. Related works - Even this section is unclear, contradictory, and confusing while some related papers are not included here but in Sect. 4. Some papers, such as [1]-[3], have already investigated RTT/delay problems in 5G applications. In addition, other works, such as [R7]-[R9], have collected experimental data, analyzed, and discussed the results about RTT/delay measurements acquired by 5G devices but they have not been cited here.

3. Test Environment Setting - The Authors are invited to specify more parameters (e.g. see Tab. 1) of the 5G setup such as frame duration, number of slots, number of symbols, modulation index, duplex mode, NR band, etc. according to 5G numerology The text portion after Tab. 1 is confused as well.

4. Statistical Analysis of 5G RTT Data - Tab. 2 is trivial and may be merged with Tab. 3. As a general note, the Authors are invited to check the x and y labels and the clarity of each figure, and table as well.

5. Time Series Analysis of 5G RTT Data - ADF test is a well-established method to analyze stationary vs non stationary properties of a sequence. I suggest the Authors to summarize this section by providing reference(s) to the method only and then presenting the results. However, if the Authors wants to describe this method, I encourage them to rewrite this section since, at the moment, it is unclear, confused and somehow incorrect. Fig. 3 is unclear as well. In addition, the order p is never estimated and the lengths of the sub-sequences are never included in the discussion and their impact never evaluated. Moreover, since the RTT sequences are non-stationary, it is not clear why a plain FFT analysis is performed. Other methods may be employed. Please discuss and rewrite.

6. 5G RTT Prediction With VMD-LSTM Method - As previously mentioned in Sect. 5 for the ADF test, since the VMD method is well-established and the description provided here is too synthetic and unclear, I suggest the Authors to summarize the mathematical description and to present only the results. Similar considerations may apply also to the LSTM method/architecture. As far as Fig. 6 is concerned, it is confused, unclear and somehow trivial: is the output block a summation or a collection of outputs? Figs. 7, 8 and 9 are unclear and uninformative as well. Moreover, the Authors are encouraged to provide an algorithmic description of their VMD-LSTSM using flow charts or pseudo-code fragments. Please check also the mathematical parts of this and other sections for missing definitions, implicit assumptions, and mistakes.

7. Results and Discussion - No comparisons against similar works are neither presented nor discussed.

Additional references

[1] Liu, Z., & Sheng, Z. (2022). URLLC occasional large time delay prediction based on unbalanced regression and LSTM. Physical Communication, 54, 101785.

[2] Li, J., Li, Z., Li, J., Shi, G., Zhang, C., & Ma, H. (2023, July). Time Series Prediction of 5G Network Data Based on Improved EEMD-BiLSTM Prediction Model. In International Conference on Intelligent Computing (pp. 409-420). Singapore: Springer Nature Singapore.

[3] Chengsong, H. (2023). Time Delay Analysis of Deterministic Network 5G-A Terminals. Advances in Computer, Signals and Systems, 7(10), 57-61.

[4] Bi, J., Ma, H., Yuan, H., & Zhang, J. (2023). Accurate Prediction of Workloads and Resources with Multi-head Attention and Hybrid LSTM for Cloud Data Centers. IEEE Transactions on Sustainable Computing.

[5] Zhou, T., Wu, W., Peng, L., Zhang, M., Li, Z., Xiong, Y., & Bai, Y. (2022). Evaluation of urban bus service reliability on variable time horizons using a hybrid deep learning method. Reliability Engineering & System Safety, 217, 108090.

[6] Wu, K., Lu, J., Lin, F., Huang, Y., Zhan, C., & Sun, L. (2023, May). A realistic network traffic forecasting method based on VMD and LSTM network. In 2023 IEEE International Symposium on Circuits and Systems (ISCAS) (pp. 1-5). IEEE.

[R7] Lackner, T., Hermann, J., Dietrich, F., Kuhn, C., Angos, M., Jooste, J. L., & Palm, D. (2022). Measurement and comparison of data rate and time delay of end-devices in licensed sub-6 GHz 5G standalone non-public networks. Procedia CIRP, 107, 1132-1137.

[R8] Rischke, J., Sossalla, P., Itting, S., Fitzek, F. H., & Reisslein, M. (2021). 5G campus networks: A first measurement study. IEEE Access, 9, 121786-121803.

[R9] Ficzere, D., Soós, G., Varga, P., & Szalay, Z. (2021, May). Real-life v2x measurement results for 5g nsa performance on a high-speed motorway. In 2021 IFIP/IEEE International Symposium on Integrated Network Management (IM) (pp. 836-841). IEEE.

Comments on the Quality of English Language

The manuscript is not easily readable mostly due to a lot of unclear and confusing parts. Some language errors and cryptical phrases make comprehension of the manuscript even more difficult. I suggest the Authors to let a mother tongue person revise the manuscript.

Round 2

Reviewer 1 Report

Comments and Suggestions for Authors

The revised paper is of good shape and can be accepted as is.

Reviewer 2 Report

Comments and Suggestions for Authors

This is my second review of the manuscript.

General remarks

The Authors have answered to most of my remarks and modified Sects. 1, 2 and 3 accordingly. However, they have introduced a lot of mathematical mistakes in the new parts of Sect. 5 and 6 that strongly limit the comprehension of the proposed method.

I have detailed below some of these mistakes (using latex-like style) of Sects. 5 and 6 but there are too many. Therefore, I suggest the Authors to completely rewrite the mathematical parts of Sect. 5 and 6 keeping in mind that every mathematical term must be defined.

Please remember also that scalar terms must be written in italic and matrices/vector in bold. It is also good practice to indicate the size of matrices and vectors. Moreover, Integer, real, and complex terms must be specified as well.

In addition, sequences (and sub-sequences) are usually described as vectors (row or column using squared brackets) or indicated using round parentheses i.e. (.,.,.). In the manuscript, there is some confusion regarding sequences/sub-sequences and elements of the sequences/sub-sequences that makes the comprehension of those parts very difficult.

Since the method is badly detailed, this fact prevents me any evaluation of the correctness of the results. In fact, the presence of mistakes and unclear parts in the mathematical description of the proposed method cast a shadow on the correctness and validity of the results. This is a pity since the results look interesting but they must be supported by a rigorous mathematical formulation.

Some specific remarks

1) Pag. 3, lines 99-100: Please explain and discuss.

2) Pag. 3, lines 102-103: This statement is not clear: network planning depends on the specific environmental conditions of the 5G area where the network is deployed. How is this obtained with the results in a specific site? It seems to me that the proposal could be used for post-planning verification only. Please clarify and discuss.

3) Tab. 1 and all tables in the manuscript: All tables take too much space. Please compact the size of all tables by deleting the white lines.

4) Pag. 9, lines 255-257: Are these datasets taken in the same site or different sites? If data i collected in the same industrial site how can you generalize the results for other industrial scenarios? Please clarify and discuss. See also the remark #2.

5) Pag. 10, line 290: T_N is never defined. In addition, I suppose that R_t belongs to the set of real positive numbers.

6) Pag. 10, line 295: I assume that RLen/n is integer. Is it true or not?

7) Pag. 10, lines 296-298: Is it RLen=50.000 and the number of the short sequences (i.e., sub-sequences) equal to 50? Please clarify.

8) Pag. 10, lines 300-301: Is p the lag order or the lag length? Please clarify.

9) Tab. 4: Please specify in the caption that the tests are performed on the sub-sequences of the original sequence {R_t}.

10) Pag. 11, line 319: Since L_{DR_t}=L{R_t}-1, where L_{DR_t} is the length of the sequence {DR_t} and L{R_t} is the length of the sequence {R_t}, not all sub-sequences of {DR_t} have the same number of samples. Please clarify and rewrite.

11) Pag. 12, lines 342-344: This text fragment is not clear. In addition, since the quality of Fig.3.b is poor it is not possible to visually inspect the plot. Please revise.

12) Fig. 3: The sub-figures a) and b) are of poor quality.

13) Pag. 13, line 367: This sequence start from element t-m while {R_{t,-m}} starts form t-m*1. Is it correct? Or not? It is not clear. Please check.

14) Pag. 13, line 373: DR_{-m} is never defined. Probably it is {DR_{t,-m}}. Please check.

15) Pag. 13, lines 373-374: DR_{t,-m,k} with k=1,...K (I suppose) is never defined. Not clear and confused. Please clarify.

16) Pag. 13, lines 375-385: This text fragment is unclear and confused. Some embedded mathematical terms are not defined. Please check and revise.

17) Pag. 13, lines 386-390: \hat{DR-{t,n,k}} is never defined. This part is confused and unclear as well. Please revise.

18) Pag. 13, line 391, eq. (2): \hat{DR-{t,n,k}} is never defined. Please revise.

19) Pag. 13, line 395, eq. (3): \Hat{DR_j} is never defined.

20) Fig. 4: This picture is too small and it is not possible to see the elements of the sequences/sub-sequences involved in the processing. Please revise.

21) Pag. 14, line 404: In Sect. 5, it was T_N while now it is t_N: are T_N and t_N the same variable? Or not? Please clarify.

22) Pag. 14, line 405: {DR_{t,k}} is never defined. It was defined {DR_{t.-m}} but {DR_{t,k}} does not make sense w.r.t. the previous definition is k is k>0. Please check and revise.

23) Pag. 14, line 413, eq. (5): Please check this equation since there is something wrong.

24) Pag. 14, line 414: \lambda_{k}^{n} is never defined. Moreover, is \lambda_t a scalar or a vector? It is confused and unclear. The same considerations apply also to the other variable. Please clarify and revise.

25) Pag. 15, line 422: \tau is never defined.

26) Pag. 15, line 424: GOA method is neither defined nor cited.

27) Pag. 15, line 426: REI method is neither defined nor cited.

28) Pag. 15, lines 435-437: this text portion is confused and unclear.

29) Pag. 16, lines 467-476: this text portion is confused and unclear as well.

30) Pag. 19, lines 527, 528: Please substitute the original and predicted sequence terms into these two equations instead of y_i and \hat{y}_i.

Comments on the Quality of English Language

The manuscript has some minor language mistakes but it is easily readable except for Sects. 5 and 6 due to the poor mathematical formulation and some obscure parts that strongly limit the comprehension of these two sections. Probably, the new revised parts have been written in a hurry.

Round 3

Reviewer 2 Report

Comments and Suggestions for Authors

My main concerns are about some hypotheses regarding the generalization of the results and lack of real requirements for URLLC industrial scenarios. Some old issues are still unsolved. For the details see the attached document.

Comments on the Quality of English Language

Some sentences are still unclear.